# Grounding LLMs in Scientific Discovery via Embodied Actions

Bo Zhang [* 1]   Jinfeng Zhou [* † 1]   Yuxuan Chen [1]   Jianing Yin [1]   Minlie Huang [1]   Hongning Wang [† 1]

## Abstract

Large Language Models (LLMs) have shown significant potential in scientific discovery but struggle to bridge the gap between theoretical reasoning and verifiable physical simulation. Existing solutions operate in a passive "execute-then-response" loop and thus lacks runtime perception, obscuring agents to transient anomalies (e.g., numerical instability or diverging oscillations). To address this limitation, we propose EmbodiedAct, a framework that transforms established scientific software into active embodied agents by grounding LLMs in embodied actions with a tight perception-execution loop. We instantiate EmbodiedAct within MATLAB and evaluate it on complex engineering design and scientific modeling tasks. Extensive experiments show that EmbodiedAct significantly outperforms existing baselines, achieving SOTA performance by ensuring satisfactory reliability and stability in long-horizon simulations and enhanced accuracy in scientific modeling. Our repository is released at https://github.com/THU-AICosmos/EmbodiedAct.

## 1. Introduction

Large Language Models (LLMs, Yang et al. 2025a) are revolutionizing scientific discovery, showing remarkable capabilities in literature review (Agarwal et al., 2024), hypothesis generation (Yang et al., 2024b; Radensky et al., 2026), and experiment automation (Chan et al., 2024). This progress is driving a paradigm shift in AI for Science, moving from data-driven pattern recognition (Gu et al., 2024) to model-driven autonomous reasoning (Zheng et al., 2025). Yet, a critical gap remains: genuine scientific discovery, particularly in process-oriented scenarios (e.g., engineering design

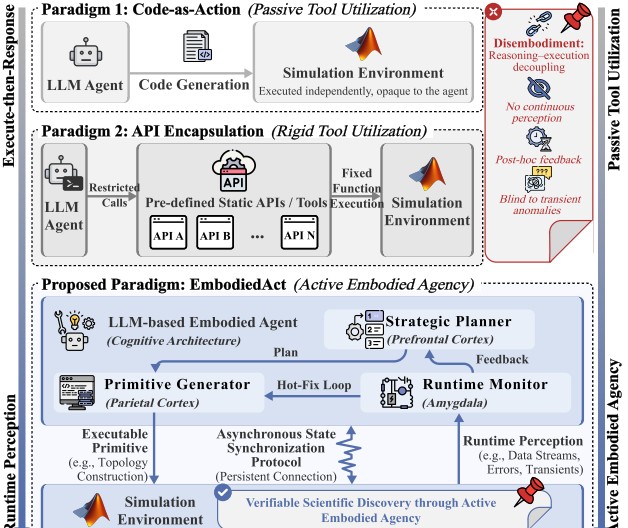

*Figure 1.* Comparison of EmbodiedAct with existing paradigms. EmbodiedAct integrates executable simulation primitives and continuous runtime perception, endowing the agent with the capacity for embodied action within physical simulation environments.

(Guo et al., 2025)), requires not just theoretical derivations, but verifiable execution within realistic environments, or via high-fidelity simulations when facing physical or other types of constraints. While LLMs excel at semantic understanding and logical deductions, their purely text-based reasoning is inherently insufficient to capture the intricate physical dynamics and real-world constraints (Lanham et al., 2023). Without the ability to ground abstract theories and reasoning with observations during executions, LLMs risk producing "hallucinated" discoveries that fail to hold up under rigorous physical verification (Li et al., 2025b; Sinha et al., 2025), leaving a divide between abstractive reasoning and verifiable scientific outcomes (Si et al., 2026).

In this work, we focus on LLM-driven scientific discovery in simulated physical environments and explore solutions to bridge the gap. Prior to ours, efforts have been made in this direction via two paradigms: **1) code-as-action**, where methods utilize LLMs to synthesize executable programs (Li et al., 2024) to realize scientific computation (Wang et al., 2024a; Ren et al., 2025); and **2) API encapsulation** (e.g., Model Context Protocols), where simulation functions within scientific software are exposed as services (Jiang et al., 2025; Yang et al., 2024a). Yet, both paradigms follow a rigid *execute-then-response* loop, where feedback is

---

[*]Equal contribution. [†]Corresponding author.

[1]Tsinghua University, Beijing, China. Correspondence to: Jinfeng Zhou <zjf23@mails.tsinghua.edu.cn>, Hongning Wang <hw-ai@tsinghua.edu.cn>.

*Proceedings of the 43rd International Conference on Machine Learning*, Seoul, South Korea. PMLR 306, 2026. Copyright 2026 by the author(s).

accessible only after execution finishes. Consequently, they lack *runtime perception* necessary to actively monitor, interpret, and intervene during the transient development of a physical process. This disembodiment obscures solutions to intermediate physical anomalies (e.g., diverging oscillations, unstable chemical reactions) that may not trigger immediate execution errors but can fundamentally invalidate scientific results and waste computational resources on doomed trials.

We believe a key barrier to verifiable discovery lies in the mechanical decoupling between discrete textual reasoning and continuous physical dynamics embedded in simulations. Bridging this gap requires more than just better code generation or richer API support, but a paradigm shift from passive tool use to *active embodied agency*.

To achieve this, we propose **EmbodiedAct**, a framework that transforms existing scientific software into active embodied agents by grounding LLMs in **Embodied Act**ion with a tight integration of execution and perception. The comparison between EmbodiedAct and existing paradigms for scientific agents is presented in Figure 1. Specifically, drawing on human cognitive architecture to support this embodiment (Anderson et al., 2004), we structure the agent's functional modules to mirror biological cognition: **1) Strategic Planner** (as prefrontal cortex) decomposes abstract scientific intent into hierarchical executive steps; **2) Primitive Generator** (as parietal cortex) translates these steps into software-specific simulation primitives (e.g., numerical solvers like `ode45` or topological block manipulation in MATLAB) to execute the operations; **3) Runtime Monitor** (as amygdala) utilizes a runtime perception engine to supervise the simulation lifecycle (*e.g., detecting latent risks, execution errors, and intermediate results*) and a reflective decision-maker to align simulation results with scientific intent for autonomous optimization. The monitor is supported by our **Asynchronous State Synchronization Protocol**, which is built on real-time web sockets to maintain a persistent connection between the LLM and scientific software, thus enabling a tight perception-action loop.

We instantiate EmbodiedAct within MATLAB, selected for its dominance in scientific modeling and graphical modeling environment of Simulink, providing an ideal testbed for this new framework. Extensive experiments confirm that equipping agents with the capacity to continuously perceive and act upon physical constraints significantly improves solution reliability, stability, and accuracy.

## 2. Related Work

LLMs are evolving from passive chatbots to autonomous agents capable of managing the entire lifecycle of research (Zheng et al., 2025), spanning literature review (Agarwal et al., 2024; Wang et al., 2024c), hypothesis generation (Yang et al., 2024b; 2025b; Hu et al., 2024; Radensky et al.,

2026), and experiment automation (Chan et al., 2024). To enhance the reliability and creativity of LLM-driven scientific discovery, recent work has advanced from controllable prompt engineering (Ciuca et al., 2023) and reinforcement learning optimization (Li et al., 2025a) to integrating knowledge retrieval (Xiong et al., 2024) and orchestrating multi-agent collaboration (Ghafarollahi & Buehler, 2024b). However, a critical gap remains between theoretical reasoning and experimental verification (Si et al., 2026).

To bridge this gap, recent work adopted the code-as-action paradigm (Gao et al., 2023; Li et al., 2024) that uses LLMs to generate executable code for verification (Jansen et al., 2025), expanding from general data science (Gu et al., 2024) to complex scientific engineering tasks (Boiko et al., 2023; Wang & Zeng, 2025). Yet, existing code-based agents follow a "generate-execute-observe" workflow (Wang et al., 2024a), where feedback is only accessible after execution terminates. This disembodiment, lacking runtime perception, is insufficient for process-oriented scientific discovery (Liu et al., 2025; Zhu et al., 2025; Koblischke et al., 2025; Ghafarollahi & Buehler, 2024a; Sinha et al., 2025), e.g., engineering design (Guo et al., 2025) and complex system prototyping (Ren et al., 2025; Liang & Zhao, 2026), where critical failures often manifest during transient evolution (e.g., voltage instability in circuit design or intermediate convergence failure in numerical optimization) rather than at the final state. Similarly, recent autonomous "AI Scientist" systems (Shao et al., 2025) used API calls to facilitate discovery (Jiang et al., 2025; Qu et al., 2025). They tend to treat the experimental environment as a black box (Krishnan, 2025), rendering them inadequate for dynamic system modeling where continuous perception is essential (Sun et al., 2024).

## 3. Methodology

As shown in Figure 2, EmbodiedAct establishes a closed-loop control architecture designed to ground scientific intent into verifiable execution. In the following, we provide our problem formulation and architecture design in details.

### 3.1. Problem Formulation

We model autonomous scientific discovery as intent-driven problem solving, which can be formulated as a sequential decision-making process within a partially observable environment $\mathcal{E}$ (exemplified by MATLAB software), which is defined by the tuple $\langle \mathcal{S}, \mathcal{A}, \mathcal{O}, \mathcal{C}, \mathcal{I} \rangle$.

- **State Space** $\mathcal{S}$ is the latent physical state of the simulation environment (e.g., *simulation primitive scripts, variable workspaces, and dynamic model structures*).
- **Action Space** $\mathcal{A}$ is the set of executable operations. An action $a_t \in \mathcal{A}$ is a synthesized primitive snippet (e.g., *invoking a solver* `ode45(...)`) or a system control command (e.g., *start_simulation, stop*).

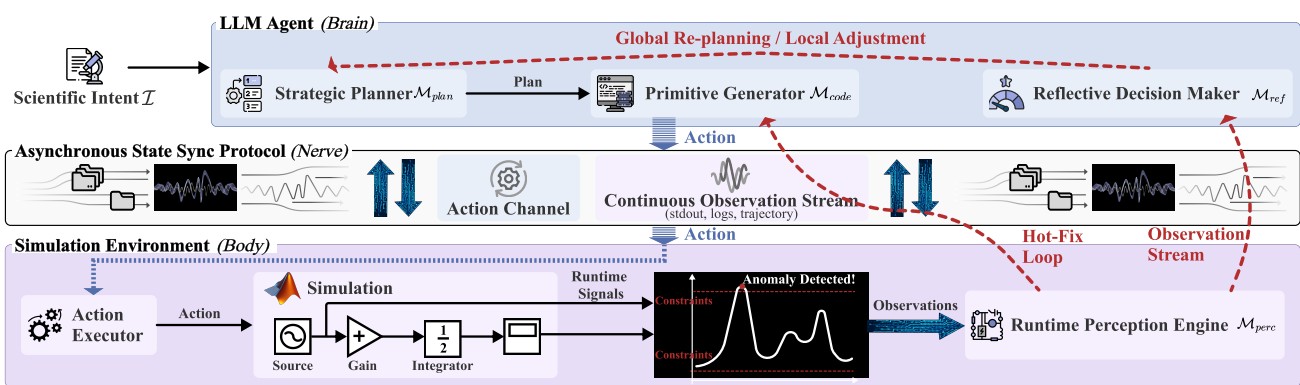

*Figure 2.* Overview of EmbodiedAct, which bridges the LLM agent and simulation environment via the Asynchronous State Sync Protocol. EmbodiedAct orchestrates a fast inner loop driven by the Runtime Perception Engine to trigger immediate Hot-Fixes, and a slow outer loop driven by the Reflective Decision Maker to guide Re-planning.

- **Observation Space** $\mathcal{O}$: Distinct from standard tool-use paradigms where observations are discrete, static text returns, we define observations as continuous streams. At any time step $t$, the agent receives a streaming observation, i.e., $o_t = \{v_{\text{stdout}}, v_{\text{stderr}}, \Psi_{\text{sys}}, \Omega_{\text{sim}}\}$, comprising standard outputs $v_{\text{stdout}}$, error logs $v_{\text{stderr}}$, system events $\Psi_{\text{sys}}$ (e.g., *warnings, interrupts*), and dynamic simulation trajectories $\Omega_{\text{sim}}$. Here, $\Omega_{\text{sim}}$ is the time-series meta-data of the physical simulation, e.g., *velocity, voltage*.
- **Scientific Intent** $\mathcal{I}$: The natural language description of the high-level scientific goal (e.g., "*Design a PID controller yielding a phase margin* $> 45°$").
- **Constraints** $\mathcal{C}$: A verification function set $\{c_1, \ldots, c_k\}$ derived from $\mathcal{I}$ (e.g., *phase margin* $> 45°$, *stable convergence*), where $c_i(s_t) \in \{0, 1\}$ is a binary indicator of whether state $s_t$ meets a specific physical constraint.

The objective of LLM-based scientific problem-solving is to synthesize a policy $\pi(a_t|o_{0:t-1}, \mathcal{I})$ that generates a trajectory $\tau = (a_0, o_0, \ldots, a_T, o_T)$ such that the final state $s_T$ satisfies all constraints: $\forall c \in \mathcal{C}, c(s_T) = 1$.

### 3.2. Architecture of the Embodied Scientific Agent

To obtain the optimal policy $\pi$ under the **EmbodiedAct** framework, we instantiate the agent with a modular cognitive architecture, including four key functions: planning, execution, perception, and reflection. Detailed implementations of these functions are provided in Appendix A.

**Strategic Planner** ($\mathcal{M}_{plan}$)  Acting as the central planner, this module bridges the gap between abstract intent and executable procedures. It decomposes the scientific intent $\mathcal{I}$ into a set of target constraints $\{c_1, \ldots, c_k\}$ and a hierarchical plan: **(1) Global planning:** Conditioned on the intent $\mathcal{I}$ and the current interaction history $\tau_{0:t}$, the planner prompts an LLM to generate a sequence of high-level sub-tasks (e.g., *Modeling* $\rightarrow$ *Simulation* $\rightarrow$ *Validation*):

$$\mathcal{P}_{global} = \{p_1, \ldots, p_n\} = \mathcal{M}_{plan,global}(\mathcal{I}, \tau_{0:t}). \quad (1)$$

**(2) Local planning:** For each sub-task $p_i$, the planner gen-

erates a detailed sequence of atomic executive steps based on real-time observations $o_{0:t}$:

$$\mathcal{P}_{local} = \{p_{i,1}, \ldots, p_{i,m}\} = \mathcal{M}_{plan,local}(p_i, o_{0:t}). \quad (2)$$

Specifically, in our MATLAB instantiation, we ground these steps in software-specific simulation primitives (e.g., `ode45, linprog` *from MATLAB Toolboxes*). By utilizing toolbox documentation as an external knowledge base, we ensure that the planned steps accurately leverage the embedded capabilities of the scientific software.

**Primitive Generator** ($\mathcal{M}_{code}$)  This module translates a logical sub-step $p_{i,j}$ into a software-specific executable primitive action $a_t$ within the simulation environment:

$$a_t = \mathcal{M}_{code}(p_{i,j}, s_t). \quad (3)$$

The generated action utilizes software-specific simulation primitives to directly manipulate the simulation environment. A critical capability of this module is **topological reasoning**. Unlike generic code generation (Li et al., 2024), $\mathcal{M}_{code}$ is equipped to understand spatial semantics within graphical environments (e.g., Simulink). It maps logical intent (e.g., "*connect the controller output to the plant input*") into executable 2D topological structures, effectively manipulating the graph-based simulation environment.

**Runtime Perception Engine** ($\mathcal{M}_{perc}$)  To enable the "active embodiment" central to EmbodiedAct, this module provides real-time latent state inference $z_t$ by processing the continuous observation stream $o_t$ and the executing action $a_t$ via our Embodied Interaction Protocol (§3.3):

$$z_t = \mathcal{M}_{perc}(a_t, o_t) \in \{\text{Normal}, \text{Error}, \text{Warning}\}, \quad (4)$$

This engine goes beyond simple syntax checking. Crucially, when $z_t$ is Normal, the module focuses on the constraint set $\{c_1, \ldots, c_k\}$. It actively monitors and records the optimal parameters in environment state $s_t$ and intermediate results during the simulation process. This is vital for long-horizon

simulations (e.g., *Finite Element Analysis in engineering design*), where capturing transient optimality is as important as the final result. Additionally, the module parses execution streams to identify potential risks, such as numerical instability (e.g., `Inf/NaN` *divergence*), algebraic loops, or stiffness warnings. Upon detecting a constraint violation or an immediate execution error, it triggers a *Hot-Fix Loop*:

$$a_t' \leftarrow \text{Repair}(a_t, o_t). \tag{5}$$

This mechanism allows the agent to not only iteratively correct syntax or runtime errors (e.g., *dimension mismatches*), but also drive autonomous parameter tuning to explore for better results without altering the high-level plan, thereby ensuring atomic executability.

**Reflective Decision Maker ($\mathcal{M}_{ref}$)** While the Perception Engine ensures runtime stability, this module evaluates goal alignment. It employs an LLM-as-a-judge mechanism (Gu et al., 2025) to verify whether the outcome $o_t$ satisfies the physical constraints $\mathcal{C}$:

$$\text{Feedback}_t = \mathcal{M}_{ref}(o_t, \mathcal{C}). \tag{6}$$

Upon detecting a failure where the set of unsatisfied constraints $\Delta = \{c_j \mid c_j(s_t) = 0\} \neq \emptyset$, the module creates a physics-informed update to enter a re-planning cycle:

$$p' \leftarrow \mathcal{M}_{plan}(p, \Delta, \text{Feedback}_t). \tag{7}$$

Depending on the nature of the error, the cycle activates one of two self-correction strategies: **(1) Local Adjustment:** If the plan is correct but performance is suboptimal (e.g., "*Overshoot is 15%, but target is $< 10\%$*"), $\mathcal{M}_{ref}$ triggers a local refinement of parameters within the current sub-task $p_i$. **(2) Global Re-planning:** If the failure indicates a basic methodological flaw (e.g., "*PID controller fails to stabilize the non-linear system*"), $\mathcal{M}_{ref}$ escalates the issue to $\mathcal{M}_{plan}$ to alter the global strategy (e.g., "*switch to Model Predictive Control*"), generating a new task plan $p_i'$.

### 3.3. Embodied Interaction Protocol of EmbodiedAct

To operationalize dynamic interaction between discrete tokens generated by the LLM (*the Brain*) and continuous runtime generated by the scientific software (*the Body*), we establish an **Asynchronous State Synchronization Protocol**. This protocol transforms the interaction paradigm through three key mechanisms:

- **Asynchronous Decoupling and Stateful Sessions:** The protocol defines execution as a continuous time interval $t \in [t_{start}, t_{end}]$ within a stateful session. Upon receiving an action $a_t$, the environment immediately returns a confirmation message (`operation_ack`, see Table 9 in Appendix for more message types), decoupling the command dispatch from the execution lifecycle. This allows the session to maintain persistent variable workspaces and operation history across interactions, mirroring the continuous focus of a human researcher.

- **Real-time Multi-modal Streaming:** During the execution interval, a software-side *Operation Tracker* actively pushes a synchronized observation stream to the perception engine. This stream encapsulates standard outputs ($v_{\text{stdout}}$), system events ($\Psi_{\text{sys}}$), and crucial dynamic simulation trajectories ($\Omega_{\text{sim}}$) as state update messages. This enables the agent to capture transient intermediate states $s_{t'}$ (where $t_{start} < t' < t_{end}$), allowing $\mathcal{M}_{perc}$ to inspect convergence trends or physical violations in real-time before the final result is finalized.

- **Bi-directional Active Intervention:** The full-duplex protocol empowers the agent with *active agency*. If $\mathcal{M}_{perc}$ infers a critical anomaly ($z_t = \text{Warning/Error}$) from the live stream, the agent is not forced to wait for the process to crash. Instead, it can asynchronously transmit a high-priority interrupt signal ($a_{stop}$) to immediately halt the simulation. This capability enables the *Hot-Fix Loop* (depicted in Figure 9 in Appendix), transforming the LLM from a passive tool caller into a responsive supervisor that prevents wasted computation or later system crashes.

Technical specifications of the message schema and state transitions are provided in Appendix C.

## 4. Experiments

### 4.1. Experiment Setup

**Benchmarks** We evaluate our MATLAB-instantiated EmbodiedAct agent on two distinct benchmarks: **(1) EngDesign** (Guo et al., 2025): A simulation-centric benchmark designed to assess LLMs' capabilities in performing practical engineering design tasks, spanning nine domains: *Control Systems ($\mathbb{CS}$), Structural Engineering ($\mathbb{SE}$), Signal Processing ($\mathbb{SP}$), Robotics ($\mathbb{ROB}$), Path Planning ($\mathbb{PP}$), Antenna Design ($\mathbb{AD}$), Computer Vision ($\mathbb{CV}$), Digital Design ($\mathbb{DD}$), and Miscellaneous ($\mathbb{MIS}$)*. We excluded 9 problems that require closed-source simulation software, resulting in a benchmark of 92 problems. Within this set, 45 problems specifically necessitate MATLAB simulation, while the remaining problems utilize alternative simulation software. **(2) SciBench-107:** Derived from SciBench (Wang et al., 2024b), a benchmark of collegiate-level scientific problems across *Mathematics, Chemistry, and Physics*. From the original SciBench, we initially selected a pool of problems suitable for MATLAB simulations. We then conducted a rigorous manual inspection to filter out samples containing annotation errors (e.g., incorrect ground-truth answers) or incomplete problem statements. This yielded a refined subset of 107 verified problems. Details about our data selection procedure are reported in Appendix B.1.

**Baselines and Evaluation Metrics** We compare EmbodiedAct with two classes of baseline methods: **(1) Generative Models**: GPT-5.2&5-mini, Claude-Opus-4.5&Sonnet-4.5 (Anthropic, 2025), Gemini-3-Pro&Flash (Team et al., 2025), Qwen3-235B-Instruct&8B-VL-Instruct (Yang et al.,

Table 1. *Average Score* on the EngDesign benchmark. "Core" set consists of 45 tasks requiring MATLAB simulation. "Extended" (Ext) set expands this to 92 tasks, employing an execution backend compatible with open-source simulation software beyond just MATLAB.

| Models | $\mathbb{CS}$ Core/Ext | $\mathbb{SE}$ Core/Ext | $\mathbb{SP}$ Core/Ext | $\mathbb{ROB}$ Core/Ext | $\mathbb{PP}$ Core/Ext | $\mathbb{AD}$ Core/Ext | $\mathbb{CV}$ Core/Ext | $\mathbb{DD}$ Core/Ext | $\mathbb{MIS}$ Core/Ext | Overall Core/Ext |
|---|---|---|---|---|---|---|---|---|---|---|
| **Generative Models** | | | | | | | | | | |
| Qwen3-8B-VL-Instruct | 18.0 / 18.0 | 14.2 / 18.3 | **80.0** / 24.9 | 13.3 / 22.2 | 45.4 / 45.4 | 13.3 / 13.3 | 15.0 / 40.1 | − / 44.3 | 26.7 / 27.5 | 21.7 / 25.9 |
| Qwen3-235B-Instruct | 33.0 / 33.0 | 28.5 / 30.5 | **80.0** / 28.9 | 13.3 / 32.6 | 68.8 / 68.8 | 13.3 / 13.3 | 53.3 / 55.2 | − / 56.7 | 78.9 / 53.1 | 39.6 / 42.4 |
| GPT-5-mini | 38.3 / 38.3 | 56.5 / 36.3 | **80.0** / 36.4 | 20.0 / 26.7 | 25.0 / 25.0 | 40.0 / 40.0 | 76.7 / 82.6 | − / 85.2 | 81.1 / 41.0 | 42.0 / 46.9 |
| Claude-Sonnet-4.5 | 38.2 / 38.2 | 25.7 / 23.4 | **80.0** / 40.0 | 13.3 / 32.3 | 68.3 / 68.3 | 33.3 / 33.3 | 40.0 / 63.3 | − / 65.9 | 72.2 / 50.1 | 42.1 / 44.5 |
| Claude-Opus-4.5 | 41.6 / 41.6 | 36.9 / 31.1 | **80.0** / 51.2 | 13.3 / 31.2 | 64.8 / 64.8 | 40.0 / 40.0 | 35.0 / 55.3 | − / 82.8 | 86.7 / 52.3 | 45.6 / 48.5 |
| GPT-5.2 | 40.3 / 40.3 | 68.2 / 41.8 | **80.0** / 29.4 | 20.0 / 37.0 | 52.7 / 52.7 | 40.0 / 40.0 | 73.3 / 76.9 | − / 84.1 | 90.0 / 61.6 | 48.0 / 51.9 |
| Gemini-3-Flash | 44.2 / 44.2 | 48.1 / 36.4 | **80.0** / 34.4 | 20.0 / 37.0 | 64.4 / 64.4 | 40.0 / 40.0 | 41.7 / 67.9 | − / 89.4 | 83.3 / 50.7 | 48.4 / 50.5 |
| Gemini-3-Pro | 44.1 / 44.1 | 33.3 / 27.0 | **80.0** / 35.0 | 20.0 / 41.3 | 83.5 / 83.5 | 40.0 / 40.0 | 53.3 / 72.6 | − / 85.6 | 90.0 / 61.5 | 50.0 / 53.2 |
| **CodeAct** | | | | | | | | | | |
| Qwen3-8B-VL-Instruct | 28.9 / 28.9 | 52.3 / 35.0 | **80.0** / 31.5 | 13.3 / 33.8 | 71.2 / 71.2 | 40.0 / 40.0 | 70.0 / 61.7 | − / 60.2 | 61.1 / 34.0 | 38.9 / 38.1 |
| Qwen3-235B-Instruct | 36.0 / 36.0 | 53.1 / 33.6 | **80.0** / 26.9 | 20.0 / 29.0 | 66.7 / 66.7 | 40.0 / 40.0 | 80.0 / 77.5 | − / 62.2 | 72.2 / 34.3 | 44.6 / 40.6 |
| Claude-Sonnet-4.5 | 44.4 / 44.4 | 54.8 / 36.1 | **80.0** / 32.7 | 20.0 / 25.1 | 71.5 / 71.5 | 46.7 / 46.7 | 50.0 / 54.9 | − / 61.1 | 87.8 / 50.6 | 50.4 / 46.5 |
| Gemini-3-Flash | 49.8 / 49.8 | 33.9 / 32.2 | **80.0** / 29.0 | 20.0 / 34.7 | 73.1 / 73.1 | 46.7 / 46.7 | 40.0 / 67.3 | − / **96.7** | 88.9 / 45.1 | 52.2 / 51.2 |
| GPT-5-mini | 45.3 / 45.3 | 69.9 / 31.6 | 73.3 / 29.9 | 16.7 / 27.5 | 77.5 / 77.5 | 60.0 / 60.0 | 86.7 / 78.6 | − / 61.9 | 70.0 / 41.1 | 52.9 / 45.7 |
| GPT-5.2 | 50.3 / 50.3 | 74.6 / 40.7 | **80.0** / 35.5 | 20.0 / 35.1 | 65.0 / 65.0 | 46.7 / 46.7 | 75.0 / 80.4 | − / 60.9 | 77.8 / 45.2 | 55.4 / 49.4 |
| Claude-Opus-4.5 | 53.9 / 53.9 | 47.6 / 33.6 | **80.0** / 39.0 | 10.0 / 33.7 | 74.8 / 74.8 | 40.0 / 40.0 | 46.7 / 52.8 | − / 72.6 | 90.0 / 47.9 | 55.7 / 50.5 |
| Gemini-3-Pro | 55.5 / 55.5 | 42.8 / 30.8 | **80.0** / 37.3 | 20.0 / 34.1 | 84.6 / 84.6 | 40.0 / 40.0 | 75.0 / 82.9 | − / 85.6 | 90.0 / 58.4 | 59.0 / 56.9 |
| **EmbodiedAct** | | | | | | | | | | |
| Qwen3-8B-VL-Instruct | 42.9 / 40.9 | 18.6 / 18.6 | 50.0 / 47.5 | 20.0 / 18.0 | 50.0 / 46.7 | 26.7 / 26.7 | 40.0 / 61.3 | − / 73.9 | 74.8 / 42.6 | 40.8 / 44.0 |
| Gemini-3-Flash | 62.6 / 64.2 | 45.3 / 45.3 | 57.1 / **53.8** | 24.0 / 25.5 | 50.0 / 33.3 | **65.7 / 65.7** | 20.0 / 63.6 | − / 85.6 | 74.8 / 49.4 | 55.2 / 55.4 |
| Qwen3-235B-Instruct | 53.4 / 53.4 | 65.3 / 40.3 | **80.0** / 31.9 | 13.3 / **60.1** | **100.0 / 100.0** | 26.7 / 26.7 | 51.7 / 68.2 | − / 61.5 | 90.0 / 60.4 | 57.0 / 55.5 |
| GPT-5-mini | 78.5 / 75.7 | 28.7 / 28.7 | 54.3 / 47.5 | 30.7 / 36.7 | 50.0 / 63.0 | 55.2 / 55.2 | 40.0 / 63.9 | − / 88.9 | 74.8 / 49.2 | 59.3 / 58.6 |
| Gemini-3-Pro | **85.0 / 82.6** | 51.7 / 51.7 | 51.4 / 47.5 | **44.0** / 34.7 | 50.0 / 63.0 | 51.6 / 51.6 | 40.0 / 69.3 | − / 93.3 | 53.8 / 46.2 | 63.7 / 61.2 |
| Claude-Sonnet-4.5 | 61.8 / 61.8 | 70.0 / 38.3 | 73.3 / 46.4 | 20.0 / 65.2 | **100.0 / 100.0** | 40.0 / 40.0 | **100.0 / 90.1** | − / 65.6 | 90.0 / 53.7 | 65.7 / 59.3 |
| Claude-Opus-4.5 | 64.1 / 64.1 | 81.0 / 46.3 | **80.0** / 50.7 | 20.0 / 58.2 | **100.0 / 100.0** | 40.0 / 40.0 | **100.0** / 86.3 | − / 83.9 | 90.0 / 59.9 | 68.2 / 63.8 |
| GPT-5.2 | 68.6 / 68.6 | **76.6 / 46.0** | **80.0** / 33.9 | 20.0 / 46.4 | **100.0 / 100.0** | 46.7 / 46.7 | 90.0 / 82.6 | − / 86.7 | **90.0 / 66.9** | **70.6 / 65.4** |

2025a), which rely solely on textual reasoning to solve scientific problems. We employ multi-modal LMs to process visual content within the problem statements. **(2) CodeAct:** (Wang et al., 2024a) a code-as-action paradigm that generates and executes code along with multi-turn interactions and iterative self-debugging. To quantify performance, we employ benchmark-specific metrics: **(1) For EngDesign:** We utilize the *Average Pass Rate*, which measures the proportion of solutions that satisfy all minimum performance specifications and constraints defined in the task description (e.g., gain margin in control systems). Additionally, we report the *Average Score* to evaluate the quality of open-ended engineering designs, where varying design parameters (e.g., response speed, power consumption, material usage) result in different performance scores. **(2) For SciBench-107:** We report *Accuracy*, defined as the percentage of test problems where the model provides the correct final answer.

### 4.2. Evaluation Results

We report the results of our extensive experimentation in Table 1 and 2, and summarize our findings in the following.

• **EmbodiedAct consistently outperforms baseline methods across diverse scientific disciplines.** As detailed in Table 1 and 2, EmbodiedAct establishes new SOTA performance. Using GPT-5.2 as the backbone, EmbodiedAct achieves an Overall Score of 70.6%/65.4% (Core/Extended) on EngDesign and an accuracy of 48.60% on SciBench-107. These results significantly surpass both the Generative Models (48.0%/51.9% and 44.86%) and CodeAct baseline

(55.4%/49.4% and 34.58%). Importantly, this performance gain is model-agnostic, consistently emerging across various model families (e.g., Claude, Qwen3), showing the universal effectiveness of our proposed embodied paradigm.

• **Active embodiment is crucial for process-oriented discovery.** The embodied paradigm excels in domains requiring dynamic verification, e.g., *Control Systems ($\mathbb{CS}$)* and *Structural Engineering ($\mathbb{SE}$)*. As shown in the Core set of Table 1, with GPT-5.2, EmbodiedAct achieves a score of 68.6% in $\mathbb{CS}$, outperforming the Generative baseline (40.3%) by more than 70% and surpassing CodeAct (50.3%) by more than 36%. Similar gains are observed in $\mathbb{SE}$. These results validate our argument that "cognitive decoupling" is a major bottleneck for existing LLM-based scientific discovery methods. While CodeAct introduces code execution for verification, it remains blind to intermediate states. In contrast, EmbodiedAct allows the agent to monitor simulation trajectories (e.g., detecting overshoot in control loops) in real-time. This enables the agent to navigate towards optimal parameters during the simulation process, effectively closing the loop between reasoning and execution.

• **Domain-specific primitives enhance scientific computation accuracy.** As shown in Table 2, the average accuracy of GPT-5.2 rises from 34.58% (CodeAct) to 48.60% (EmbodiedAct). The advantage is evident in domains where numerical stability is paramount, such as *Mathematics* (50.00% → 83.33%) and *Physics* (28.20% → 35.90%). This can be attributed to our *Primitive Generator*, which translates intent into robust, software-specific primitives (e.g., MAT-

*Table 2. Accuracy* on the SciBench-107 (%). The detailed textbook categories (e.g., `atkins`) of each subject are shown in Appendix B.1.

| Models | Chemistry | | | | | Physics | | | | Math | | | | Avg. |
|---|---|---|---|---|---|---|---|---|---|---|---|---|---|---|
| | atkins | chemmc | quan | matter | avg. | fund | class | thermo | avg. | diff | stat | calc | avg. | |
| **Generative Models** | | | | | | | | | | | | | | |
| Qwen3-8B-VL-Instruct | 12.50 | 25.00 | 12.50 | 0.00 | 12.50 | 12.50 | 9.52 | 10.00 | 10.25 | 6.25 | 10.00 | 20.00 | 11.11 | 11.21 |
| Gemini-3-Flash | 12.50 | 25.00 | 25.00 | 12.50 | 18.75 | 12.50 | 19.05 | 20.00 | 17.95 | 25.00 | 40.00 | 70.00 | 41.67 | 26.17 |
| Qwen3-235B-Instruct | 12.50 | **62.50** | 25.00 | 0.00 | 25.00 | 25.00 | 14.29 | 30.00 | 20.52 | 18.75 | 60.00 | 50.00 | 38.89 | 28.04 |
| GPT-5-mini | 25.00 | 50.00 | 50.00 | 25.00 | 37.50 | 37.50 | 19.05 | 50.00 | 30.77 | 31.25 | 70.00 | 80.00 | 55.56 | 41.12 |
| GPT-5.2 | 0.00 | 37.50 | **62.50** | 37.50 | 34.38 | 37.50 | 23.81 | 20.00 | 25.64 | 75.00 | 70.00 | 80.00 | 75.00 | 44.86 |
| Claude-Sonnet-4.5 | 12.50 | 50.00 | **62.50** | 25.00 | 37.50 | 50.00 | 23.81 | 40.00 | 33.33 | 31.25 | **90.00** | 90.00 | 63.89 | 44.86 |
| Gemini-3-Pro | 37.50 | 37.50 | 25.00 | 37.50 | 34.38 | **62.50** | 14.29 | 40.00 | 30.77 | 68.75 | 70.00 | 90.00 | 75.00 | 46.73 |
| Claude-Opus-4.5 | **62.50** | 37.50 | **62.50** | **62.50** | **56.25** | 37.50 | 42.86 | **90.00** | **53.85** | 62.50 | 30.00 | 80.00 | 58.33 | 56.07 |
| **CodeAct** | | | | | | | | | | | | | | |
| Qwen3-8B-VL-Instruct | 25.00 | 12.50 | 0.00 | 12.50 | 12.50 | 0.00 | 14.29 | 44.44 | 19.09 | 50.00 | 80.00 | 80.00 | 66.67 | 32.71 |
| Gemini-3-Flash | 25.00 | 37.50 | 0.00 | 12.50 | 18.75 | 12.50 | 14.29 | 20.00 | 15.39 | 50.00 | 80.00 | 90.00 | 69.44 | 34.58 |
| Claude-Sonnet-4.5 | 12.50 | 37.50 | 50.00 | 25.00 | 31.25 | 25.00 | 23.81 | 50.00 | 30.77 | 37.50 | 80.00 | 80.00 | 61.11 | 41.12 |
| Claude-Opus-4.5 | 12.50 | 37.50 | **62.50** | 25.00 | 34.38 | 42.86 | 33.33 | 55.56 | 40.98 | 37.50 | 80.00 | 80.00 | 61.11 | 44.86 |
| Gemini-3-Pro | 37.50 | 37.50 | 50.00 | 25.00 | 37.50 | 12.50 | 28.57 | 50.00 | 30.77 | 75.00 | **90.00** | 70.00 | 77.78 | 48.60 |
| GPT-5-mini | 25.00 | 25.00 | 37.50 | 0.00 | 21.88 | 14.29 | 38.10 | 55.56 | 37.69 | 81.25 | **90.00** | 90.00 | 86.11 | 48.60 |
| GPT-5.2 | 25.00 | 25.00 | 37.50 | 25.00 | 28.12 | 25.00 | 38.10 | 50.00 | 38.46 | 75.00 | **90.00** | 80.00 | 80.56 | 49.53 |
| Qwen3-235B-Instruct | 37.50 | 50.00 | 50.00 | 25.00 | 40.62 | 25.00 | 38.10 | 50.00 | 38.46 | 81.25 | **90.00** | 90.00 | 86.11 | 55.14 |
| **EmbodiedAct** | | | | | | | | | | | | | | |
| Qwen3-8B-VL-Instruct | 12.50 | 0.00 | 0.00 | 25.00 | 9.38 | 14.29 | 14.29 | 33.33 | 19.17 | 31.25 | 80.00 | 50.00 | 50.00 | 26.17 |
| Gemini-3-Flash | 12.50 | 25.00 | 50.00 | 37.50 | 31.25 | 12.50 | 33.33 | 40.00 | 30.77 | 62.50 | **90.00** | 90.00 | 77.78 | 46.73 |
| GPT-5-mini | 12.50 | 37.50 | 37.50 | 25.00 | 28.12 | 14.29 | 42.86 | 55.56 | 40.26 | 68.75 | **90.00** | 90.00 | 80.56 | 49.53 |
| Qwen3-235B-Instruct | 37.50 | 37.50 | **62.50** | 25.00 | 40.62 | 0.00 | 33.33 | 55.56 | 32.19 | 75.00 | **90.00** | 100.00 | 86.11 | 52.34 |
| Claude-Sonnet-4.5 | 25.00 | 25.00 | **62.50** | 37.50 | 37.50 | 12.50 | 38.10 | 50.00 | 35.90 | 75.00 | 80.00 | 100.00 | 83.33 | 52.34 |
| GPT-5.2 | 25.00 | 25.00 | 50.00 | 25.00 | 31.25 | 12.50 | 47.62 | 50.00 | 41.03 | 87.50 | **90.00** | 90.00 | 88.89 | 54.21 |
| Gemini-3-Pro | 37.50 | 25.00 | 50.00 | 25.00 | 34.38 | 12.50 | **52.38** | 50.00 | 43.59 | 93.75 | 80.00 | 90.00 | 88.89 | 56.07 |
| Claude-Opus-4.5 | 25.00 | 50.00 | **62.50** | 37.50 | 43.75 | 12.50 | 52.38 | 60.00 | 46.15 | 87.50 | **90.00** | 100.00 | **91.67** | **60.75** |

LAB's optimized solvers like `ode45` or `linprog`) instead of generic, error-prone programs. As a result, the agent ensures that compute is efficiently spent on high-level scientific reasoning rather than low-level debugging.

• **Robustness across diverse simulation backend.** The "Extended" set in Table 1 evaluates the agent's adaptability to execution backend beyond standard MATLAB environments. Despite the shift to open-source simulation software, EmbodiedAct maintains its performance lead (65.4% vs. 49.4% for CodeAct with GPT-5.2). This shows that our EmbodiedAct generalizes effectively, capable of handling diverse tasks across different simulation environments.

• **Bridging the performance gap for open-source models.** While proprietary models like GPT-5.2 lead in the competition, EmbodiedAct significantly enhances open-source models. For instance, on the EngDesign Core set (Table 1), open-source Qwen3-235B-Instruct improves from 39.6% (Generative) to 57.0% (EmbodiedAct), distinctly outperforming GPT-5.2 (48.0%). This suggests that EmbodiedAct provides powerful scaffolding, compensating the raw reasoning disparities of open-source models and enabling them to perform high-capability scientific discovery.

### 4.3. Analysis of Reliability and Stability

• **EmbodiedAct shows strong reliability in multi-trial performance.** Figure 3 reports the average pass rate on the Pass@3 setting, which evaluates the agent's ability to yield a valid solution within three attempts. EmbodiedAct consistently dominates both Generative and CodeAct baselines across all model backbones. We attribute this reliability to runtime perception, which enables EmbodiedAct to detect potential errors mid-execution and triggers the *Hot-Fix Loop* to dynamically re-calibrate, thereby maximizing the success rate of each solution attempt. Crucially, the performance gap between the Core (MATLAB-centric) and Extended (open-source backend) sets is minimal in EmbodiedAct compared to the baselines. This shows EmbodiedAct can effectively generalize across different simulation engines, maintaining high reliability even in diverse execution environments.

• **EmbodiedAct significantly minimizes performance variance.** Figure 4 visualizes the variance of performance by plotting the minimum ($x$-axis) versus maximum ($y$-axis) scores across three independent runs for each task in the Extended set. Ideally, results should cluster along the diagonal ($y = x$), indicating perfect stability. Quantitative analysis reveals that EmbodiedAct achieves the highest stability: **81.5%** of its tasks fall within a narrow divergence gap (gap $\leq 20$), significantly outperforming Generative models (73.9%) and CodeAct (63.0%). Notably, there is a "Zero Zone" (pink shaded region, where $Min \approx 0$ but $Max > 0$), representing trials where models fail catastrophically in some attempts while succeeding in others. Baseline methods show a dense distribution in this zone. Conversely, EmbodiedAct effectively vacates this zone. By utilizing the Runtime Perception Engine and Hot-Fix Loop to monitor intermediate physical states, EmbodiedAct prevents divergence before it becomes irreversible, ensuring that the "worst-case" scenario (Min score) remains competitive with the "best-case" scenario.

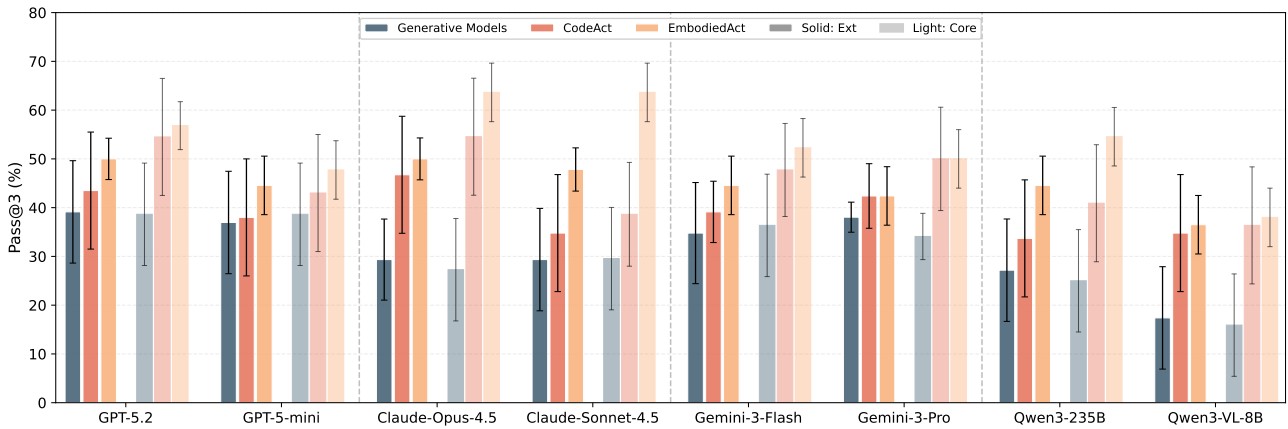

*Figure 3.* Results (%) of the average pass rate (Pass@3) on the EngDesign benchmark. A shorter error bar indicates greater reliability.

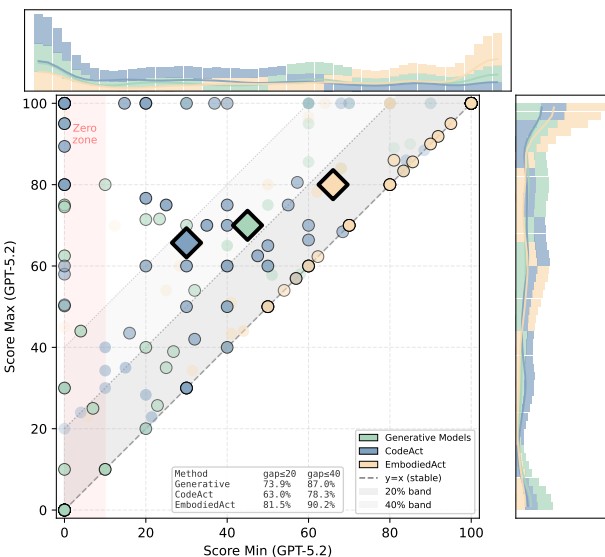

*Figure 4.* Analysis of Stability. The consistency of the minimum ($x$-axis) versus maximum ($y$-axis) scores (on the EngDesign, Extended set) across three independent runs. Ideally, results cluster along the diagonal ($y = x$), indicating perfect stability.

### 4.4. Ablation Study

We study the impact of two modules in EmbodiedAct: **(1) w/o $\mathcal{M}_{ref}$**, we disable the *Reflective Decision Maker*: The agent executes its initial plan without global re-planning, relying solely on the local Hot-Fix Loop for execution. **(2) w/o $\mathcal{M}_{perc}$**, we disable the *Runtime Perception Engine*: This removes the agent's ability to monitor runtime states (e.g., streaming trajectories) and deactivates the Hot-Fix Loop, relying solely on the post-hoc outcomes for global reflection. **(3) w/o Both**: The agent degenerates into a one-shot generator of simulation primitives, akin to a static API call. We evaluated these variants on EngDesign (Core set) and SciBench-107 using GPT-5.2 and GPT-5-minis. The results presented in Table 3 support two conclusions.

• **Runtime perception is the primary driver of verifiable discovery.** Results show that removing the perception

*Table 3.* Ablation results of average score on EngDesign (Core set) and accuracy (%) on SciBench-107. w/o is the ablated variant.

| Models | GPT-5.2 | | GPT-5-mini | |
|---|---|---|---|---|
| | EngDesign | SciBench-107 | EngDesign | SciBench-107 |
| Generative Models | 48.0 | 44.9 | 42.0 | 41.1 |
| CodeAct | 55.4 | 49.5 | 52.9 | 48.6 |
| **EmbodiedAct** | **70.6** | **54.2** | **59.3** | **49.5** |
| w/o $\mathcal{M}_{ref}$ | 65.0 | 51.0 | 55.0 | 48.0 |
| w/o $\mathcal{M}_{perc}$ | 56.5 | 50.0 | 48.5 | 46.5 |
| w/o Both | 51.5 | 47.5 | 45.5 | 45.0 |

engine ($\mathcal{M}_{perc}$) consistently causes a far more drastic performance drop than removing the reflective planner ($\mathcal{M}_{ref}$). This evidence supports our hypothesis that seeing the execution process is more critical than reflecting on the result. Without $\mathcal{M}_{perc}$, the agent loses the ability to distinguish between a theoretically correct script and a physically diverging simulation, rendering high-level planning ineffective.

• **Active embodiment distinguishes EmbodiedAct from CodeAct.** A vital observation is that the **w/o $\mathcal{M}_{perc}$** variant performs comparably to the **CodeAct** (e.g., 56.5 vs. 55.4 on EngDesign, GPT-5.2). This implies that merely switching tools, i.e., from generic code (CodeAct) to domain-specific simulation primitives (EmbodiedAct), provides slight gains if the agent remains disembodied. The substantial leap is unlocked only when the tool use is coupled with active runtime perception. Moreover, the **w/o Both** setting (51.5) falls significantly below CodeAct (55.4), confirming that static tool invocation without iterative debugging (whether textual or physical) is insufficient for complex engineering tasks. Thus, the superiority of EmbodiedAct stems not from better tools, but from the closed-loop cognitive architecture that bridges execution and perception.

### 4.5. Case Study: Magnetic Levitation Control

We conduct a qualitative analysis on a PID controller design task for a third-order magnetic levitation system (see Figure 5). The objective is to synthesize PID parameters $(K_p, K_i, K_d)$ that satisfy five rigorous coupled constraints: settling time $T_s < 5s$, overshoot $M_p < 20\%$, steady-state

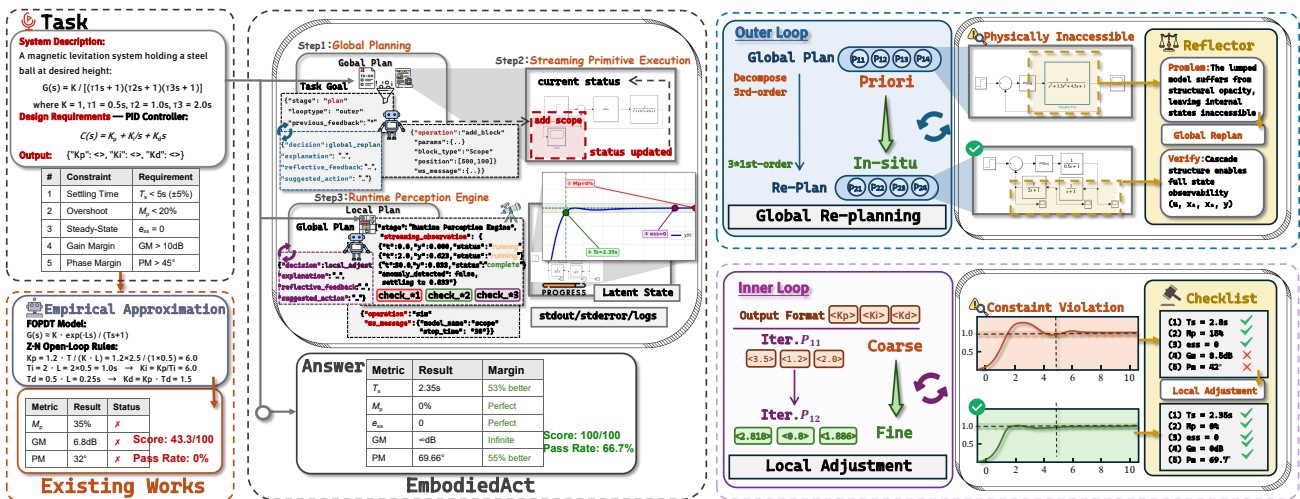

*Figure 5.* **Case Study: PID Controller Design for Magnetic Levitation. (Left)** Traditional static methods (e.g., Ziegler-Nichols tuning on FOPDT approximation). **(Right)** EmbodiedAct succeeds via a **dual-loop cognitive architecture**: the *Outer Loop* (Top Right) performs structural replanning to resolve opacity, while the *Inner Loop* (Bottom Right) executes physics-informed parameter tuning.

error $e_{ss} = 0$, gain margin $GM > 10$dB, and phase margin $PM > 45°$.

**Failure of Static Approximation**    Traditional methods relying on static theoretical deduction fail significantly on this task (Pass Rate: 0%). As analyzed in Figure 5 (Left), the classical Ziegler-Nichols (Z-N) method approximates the high-order plant as a First-Order Plus Dead Time (FOPDT) model. This approximation compresses the system's actual $-270°$ phase lag into a simplified $-90°$ lag plus delay, leading to severe phase mismatch at the crossover frequency. Thus, the Z-N tuned parameters yield a phase margin of only $32°$ (violating the $> 45°$ constraint) and an overshoot of $35\%$ (violating $< 20\%$), highlighting the inadequacy of disembodied theoretical derivation for complex dynamics.

**Dual-Loop Optimization of EmbodiedAct**    In contrast, EmbodiedAct achieves a 100% success rate by leveraging its dual-loop architecture to navigate the solution space dynamically. *1) Outer Loop:* In the initial phase, the *Reflective Decision Maker* detects a structural flaw: a simple P-controller fails to eliminate steady-state error ($e_{ss} \neq 0$) for this type-0 system. This triggers the outer loop, where the planner reconstructs the topology into a PID structure. Crucially, to overcome the "structural opacity" of the lumped transfer function where internal states are inaccessible, the agent autonomously decomposes the plant into three cascaded first-order blocks (Figure 5, Bottom Right). This topological restructuring enables the *Runtime Perception Engine* to directly monitor intermediate states $(x_1, x_2)$ via scopes, granting full state observability for precise fault localization. *2) Inner Loop:* With topology fixed, the agent applies physics-informed reasoning by reducing the loop gain lowers the crossover frequency, thereby recovering phase margin. In the second iteration, the agent executes a "Local Adjustment", refining param-

eters to ($K_p = 2.818, K_i = 0.8, K_d = 1.886$). This fine-tuning satisfies all five constraints with comfortable margins ($GM = \infty, PM = 69.66°$), validating the efficacy of EmbodiedAct's closed-loop cognitive architecture.

## 5. Conclusions

In this paper, we introduced EmbodiedAct, a framework designed to bridge the gap between theoretical reasoning and verifiable execution in simulation-based scientific discovery. EmbodiedAct empowers LLMs with continuous runtime perception and the ability to intervene during transient simulation states. We instantiated EmbodiedAct within MATLAB and showed its strong potential in handling complex engineering designs and scientific modeling tasks. Extensive experiments confirm that equipping agents with the capacity to continuously perceive and act upon physical constraints significantly improves solution reliability, stability, and accuracy. We believe this work marks a pivotal step toward verifiable autonomous scientific discovery, suggesting that future AI scientists should not only reason about the actions but also actively navigate through the dynamic physical environments to ground their actions and plans.

One immediate and important future direction to pursue is leveraging the base model's multi-modal perception and reasoning ability during the execution of problem-solving. Currently the model's multi-modal ability is only used to understand the scientific intent, and utilizing it to handle the environment's multi-modal feedback should further boost the framework's scope in addressing real-world problems. Another promising direction is to equip different modules in EmbodiedAct with different models, e.g., GPT for planning and Claude for primitive generation, which has the potential to create a synergy among different models' strength.

## Impact Statement

This work introduces EmbodiedAct, a framework designed to transform Large Language Models (LLMs) from passive tool users into active, embodied agents capable of verifiable scientific discovery and engineering design. By grounding abstract reasoning in high-fidelity simulations (e.g., MAT-LAB/Simulink) with continuous runtime perception, our approach significantly improves the reliability and accuracy of automated scientific workflows.

**Accelerating Scientific and Engineering Progress** The primary positive impact of this research is the acceleration of scientific discovery and engineering innovation. By automating complex, process-oriented tasks—such as control system design or dynamic modeling—EmbodiedAct lowers the barrier to entry for utilizing sophisticated scientific software. This democratization allows researchers and engineers to rapidly prototype ideas, verify hypotheses, and optimize designs without needing mastery of low-level implementation details. This could lead to faster breakthroughs in fields ranging from robotics to material science.

**Reliability, Safety, and the Simulation-to-Reality Gap** While EmbodiedAct enhances reliability within simulation environments by catching transient errors (e.g., numerical instability), relying on simulation-based verification introduces specific risks. There is a danger of "simulation exploitation," where an agent optimizes a design to satisfy constraints within the imperfections or specific boundaries of the simulator, resulting in solutions that fail or are unsafe in the real world. Users must remain aware that verifiable execution in simulation is not equivalent to physical validation. Furthermore, as agents become more autonomous, there is a risk of automation bias, where human operators may over-trust the agent's "verified" outputs without scrutinizing the underlying physics or modeling assumptions.

**Dual-Use and Misuse Potential** As with any powerful automated engineering tool, there is a potential for dual-use. The same capabilities that allow an agent to design stable magnetic levitation systems or optimize flight trajectories could theoretically be repurposed by malicious actors to design components for harmful systems (e.g., autonomous weapon) with greater ease. While our current focus is on benign scientific benchmarks, the broader deployment of such "AI Scientists" necessitates the development of safety guardrails—potentially integrated into the agent's Strategic Planner—to detect and refuse requests that violate safety or ethical guidelines.

**Economic Considerations** Automating high-level engineering tasks may impact the labor market for specialized simulation engineers. We anticipate a shift where human expertise moves towards defining scientific intents and performing final real-world validation, while the agent handles the iterative computational labor.

## Acknowledgements

This work was supported in part by the Major Research Plan of the National Natural Science Foundation of China (Grant No. 92570203) and the Beijing Natural Science Foundation (Grant No. Z250001).

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

# A. Prompts for Embodied Scientific Agent

This appendix documents the prompt design for the four-module cognitive architecture (§3.3). We present condensed templates highlighting key constraints; **complete prompts are available in our anonymous repository**.[1]

**Design Principles.** Our prompts adhere to three principles to ensure fair evaluation: **(1) Zero task-specific knowledge**: No few-shot examples, problem IDs, or ground-truth signals are embedded. **(2) Generic capability declaration**: Toolbox references (e.g., `ode45`, `fsolve`) describe general MATLAB functions, not problem-specific hints. **(3) Uniform instantiation**: All modules receive identical prompt templates across all benchmark tasks in our evaluation.

## A.1. Module Specifications

Table 4 maps each module to its formal definition in §3.2 and summarizes key prompt constraints.

| Module | Output | Key Prompt Constraints |
|---|---|---|
| $\mathcal{M}_{plan}$ | $\mathcal{P}_{global}, \mathcal{P}_{local}$ | Hierarchical decomposition; toolbox-grounded steps; first-principles derivation; validation checkpoints |
| $\mathcal{M}_{code}$ | $a_t$ | Executable primitives only; intermediate variable logging; ASCII-only syntax; scope-safe closures |
| $\mathcal{M}_{perc}$ | $z_t \in \{N, E, W\}$ | Tri-state classification (Normal/Error/Warning); streaming observation parsing; Hot-Fix trigger |
| $\mathcal{M}_{ref}$ | Feedback$_t$, $p'$ | LLM-as-judge verification; local adjustment vs. global re-planning; physics-informed error analysis |

*Table 4.* Four-module prompt architecture aligned with §3.2 formalism. N/E/W denotes Normal/Error/Warning states.

## A.2. Core Prompt Directives

Below we present the essential directives for each module (abridged for space).

---

**Strategic Planner** ($\mathcal{M}_{plan}$)                                                                 *Central Executive*

```
You are a scientific problem solver.  Decompose the problem hierarchically:
GLOBAL PLAN: High-level sub-tasks {p_1,...,p_n} (Modeling -> Simulation -> Validation)
LOCAL PLAN: Atomic steps {p_i,1,...,p_i,m} grounded in MATLAB toolbox primitives
CONSTRAINTS: (1) Use toolbox functions (ode45, fsolve, linprog)---never hand-code
numerical methods; (2) Derive formulas from first principles; (3) Include sanity
checks for each sub-task; (4) Output structured JSON with problem_type, equations,
compute_steps.
```

---

**Primitive Generator** ($\mathcal{M}_{code}$)                                                                 *Code Synthesizer*

```
Translate plan step p_i,j into executable MATLAB primitive a_t.
CONSTRAINTS: (1) ASCII-only code; (2) Log intermediates via disp(['INTERMEDIATE:',
num2str(val,15)]); (3) Assign final answer to result; (4) Scope-safety:  pass external
values as function arguments.
For topological operations:  map logical intent to 2D block coordinates (Simulink
mode).
```

---

**Runtime Perception Engine** ($\mathcal{M}_{perc}$)                                                           *State Monitor*

```
Process observation stream o_t = {v_stdout, v_stderr, Psi_sys, Omega_sim} and infer state:
OUTPUT z_t:  NORMAL (execution success) | ERROR (fatal) | WARNING (recoverable risk)
TASKS: (1) Parse stdout/stderr for results; (2) Detect numerical instability (Inf/NaN);
(3) Extract intermediate variables; (4) On WARNING/ERROR: trigger Hot-Fix Loop a'_t    <-
Repair(a_t, o_t).
```

---

[1] https://anonymous.4open.science/r/EmbodiedAct

---

**Reflective Decision Maker ($\mathcal{M}_{ref}$)** *Constraint Verifier*

```
Verify outcome o_t against constraint set C = {c_1,...,c_k}.
RECOVERY STRATEGIES:
-- Local Adjustment:  Parameter tuning within current plan (e.g., refine tolerance)
-- Global Re-planning:  Escalate to M_plan for methodology change (e.g., switch solver)
OUTPUT: Feedback_t with unsatisfied constraints Δ and recommended strategy.
```

---

### A.3. Baseline Configuration: CodeAct

For fair comparison, we implement CodeAct (Wang et al., 2024a) following its original Python-based paradigm while ensuring equivalent computational capabilities:

- **Execution backend**: Python interpreter with `matlab.engine` integration, enabling direct calls to MATLAB toolbox functions (e.g., `eng.ode45()`, `eng.fsolve()`, `eng.hinfsyn()`)
- **Tool parity**: Identical MATLAB primitives are accessible via the Python-MATLAB bridge, ensuring no computational advantage from tool availability
- **Prompt structure**: ReAct-style with `<thought>`...`</thought>` and `<execute>`...`</execute>` tags
- **Interaction budget**: Maximum 5 turns per problem (identical to EmbodiedAct)
- **Observation**: Post-execution stdout/stderr only (no streaming, no intermediate states)

The key difference is architectural: CodeAct operates as a single-prompt agent with post-hoc observation, while EmbodiedAct employs a four-module cognitive architecture with real-time streaming perception.

### A.4. Experimental Fairness

We ensure controlled comparison through the following measures:

*Table 5.* Controlled experimental setup. Both methods access identical MATLAB capabilities; the architectural difference—streaming perception and modular cognition—is the sole variable under evaluation.

| Control Variable | CodeAct | EmbodiedAct |
|---|---|---|
| LLM backbone | Identical (GPT-5.2 / Claude-4.5 / Qwen3) | |
| Max interaction turns | 5 | 5 |
| Problem input | Identical statements, no ground-truth | |
| Task-specific tuning | None | None |
| MATLAB toolbox access | Equivalent (Python bridge / native) | |
| *Differentiator* | Single-prompt ReAct | Four-module architecture |
| | Python + matlab.engine | Native MATLAB runtime |
| | Post-hoc feedback | Streaming observation |
| | — | Hot-Fix Loop |

## B. Experiments

### B.1. SciBench-107 Subset Selection

SciBench (Wang et al., 2024b) is a college-level scientific problem-solving benchmark that **comprises 580 problems** drawn from 10 undergraduate STEM textbooks that span physics, chemistry, mathematics, and engineering (Table 6). We selected this benchmark because its problems require multi-step numerical reasoning, iterative approximation, and differential equation solving, where closed-form pattern matching fails, and first-principles constraint modeling becomes necessary.

A key challenge for method development is that frontier LLMs already achieve strong performance on the full benchmark. Our comprehensive evaluation of GPT-5.2-chat on all 580 problems using zero-shot prompting yielded 75.2% accuracy (437/580, 95% CI: [71.5%, 78.6%]), leaving limited headroom for demonstrating improvements on the overall metric. We believe that systematically analyzing, classifying and addressing persistent failures of the most capable LLMs is valuable to understanding their fundamental limitations and guiding method development.

**Human Expert Error Analysis.** To characterize these persistent failures, we extracted all the problems in which GPT-5.2 produced incorrect answers and conducted an exhaustive error analysis with *domain experts* (graduate students in physics

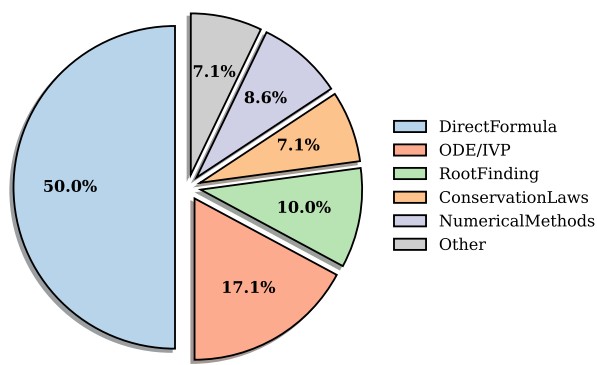

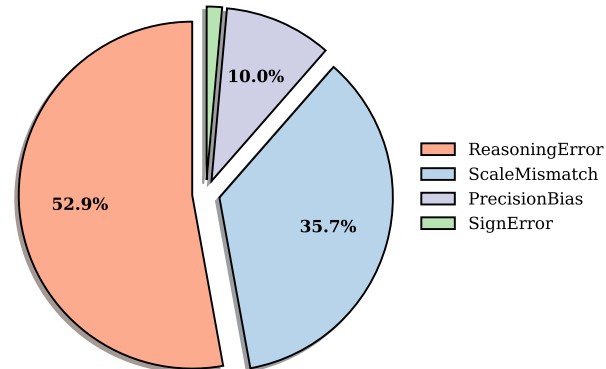

*Figure 6.* Computation pattern distribution among GPT-5.2 baseline failures. Problems requiring iterative numerical methods (ODE solving, root-finding) constitute over 25% of failures.

*Figure 7.* Error cause distribution based on human expert analysis. ScaleMismatch (35.7%) indicates systematic unit/exponent handling errors that tool-augmented computation can address.

and applied mathematics). Each of the error cases was independently reviewed and annotated along two dimensions by at least two annotators, with disagreements resolved through discussion. Inter-annotator agreement was $\kappa = 0.78$ (substantial agreement) before discussion; 12% of cases required discussion to resolve.

For *the computation pattern*, experts categorized each problem according to its underlying mathematical structure (Figure 6): application of the direct formula (50.0%), ODE/IVP solving (17.1%), root-finding (10.0%), conservation law constraints (7.1%) and other numerical methods (8.6%). For *the cause* of error, we developed a four-category taxonomy based on a systematic comparison between the model predictions and the ground truth (Figure 7):

- **ReasoningError** (52.9%): Conceptual misunderstanding, formula misapplication, or answer extraction failures.

- **ScaleMismatch** (35.7%): Predictions differing from ground truth by more than $100\times$ (e.g., $2.6 \times 10^{-10}$ vs 2.6), indicating systematic unit or exponent handling errors. The $100\times$ threshold was chosen to distinguish unit/exponent errors from numerical precision issues.

- **PrecisionBias** (10.0%): Near-correct responses with 5 to 20% relative error, suggesting a correct approach but accumulated numerical imprecision.

- **SignError** (1.4%): Correct magnitude but opposite sign.

The high prevalence of ScaleMismatch errors (35.7%) is particularly noteworthy. These represent cases where the LLM's reasoning was directionally correct but failed at maintaining proper units or exponents through multi-step calculations, precisely the type of error that tool-augmented computation can address.

**Representative Case Studies.** We present three representative cases illustrating our annotation methodology.

---

**Case B.1: Computation Pattern—ODE/IVP with Event Detection** `diff_2.3.35`

**Problem.** *IVP:* $y' = ty(4 - y)/3$, $y(0) = 0.5$. *Find $T$ such that $y(T) = 3.98$.*

**Pattern Classification.** This problem requires solving an initial value problem with event detection—the solver must integrate forward in time until $y(t)$ crosses the threshold 3.98. This cannot be solved by direct formula application; it requires numerical ODE integration with root-finding at the boundary.

**Expert Annotation.** Classified as `ODE/IVP` (16.9% of failures). The baseline model attempted symbolic integration, producing an incorrect closed-form answer instead of using numerical methods.

---

---

**Case B.2: Error Cause—ScaleMismatch (Unit Conversion Error)** `class_2.6B`

**Problem.** *A ball is thrown upward with initial velocity $v_0 = 20$ m/s. Find the time when the ball returns to the ground.*

**Model Output.** $t = 4.04$ s

**Ground Truth.** $t = 0.68$ s

**Expert Analysis.** The model's prediction differs from ground truth by $\sim 6\times$. Inspection reveals the model used $g = 1.63$ m/s$^2$ (lunar gravity) instead of $g = 9.8$ m/s$^2$ (Earth gravity), possibly confused by problem context mentioning "gravity." Classified as **ScaleMismatch**—the reasoning approach was correct but the physical constant was wrong by an order of magnitude.

---

**Case B.3: Benchmark Quality Issue—Incorrect Ground Truth** `stat_1.4.5`

**Problem.** *Given $P(A) = 0.8$, $P(B) = 0.5$, $P(A \cup B) = 0.9$, find $P(A \cap B)$.*

**Ground Truth.** $P(A \cap B) = 0.9$

**Model Output.** $P(A \cap B) = 0.4$

**Analysis.** By the inclusion-exclusion principle: $P(A \cap B) = P(A) + P(B) - P(A \cup B) = 0.8 + 0.5 - 0.9 = 0.4$ The ground truth 0.9 violates probability axioms since $P(A \cap B) \leq \min(P(A), P(B)) = 0.5$. The model's answer is mathematically correct; the benchmark annotation is erroneous. This case was retained in our subset to preserve the natural error distribution, but we flag it as a **benchmark quality issue** rather than a model failure. For evaluation metrics, we report results with these 3 identified benchmark quality issues included; excluding them would increase reported accuracy by $\sim 2.8\%$.

**Benchmark Quality Observations.** During our expert review, we identified several quality issues in the original SciBench dataset that affect evaluation reliability:

- **Incomplete problem statements**: Some problems reference parameters from preceding questions without including them (e.g., `class_2.18B` states "Include air resistance...in the previous problem" but does not provide the referenced parameters).

- **Missing domain constants**: Certain problems require specialized constants not provided in the problem text (e.g., `atkins_p2.45(b)` requires the Joule-Thomson coefficient $\mu$ which varies significantly across refrigerant types).

- **Ambiguous ground truth**: A small number of problems have questionable reference answers (see Case B.3 above).

- **Unit ambiguity**: Some problems do not clearly specify units (e.g., `quan_17.9` where the force constant units affect whether the answer is 27 or 113 kJ/mol).

These observations informed our subset construction: we retained representative failure cases to preserve the natural error distribution, while acknowledging that a portion of "failures" may reflect benchmark quality issues rather than model limitations.

*Table 6.* SciBench subject distribution showing original benchmark and our curated subset.

| Textbook | Domain | Original | Subset |
|----------|--------|----------|--------|
| class | Classical Mechanics | 56 | 21 |
| diff | Differential Equations | 50 | 16 |
| thermo | Thermodynamics | 66 | 10 |
| stat | Statistics | 72 | 10 |
| calculus | Calculus | 42 | 10 |
| quan | Quantum Mechanics | 33 | 10 |
| atkins | Physical Chemistry | 105 | 8 |
| fund | Fundamental Physics | 71 | 8 |
| matter | Materials Science | 47 | 8 |
| chemmc | Quantum Chemistry | 38 | 6 |
| **Total** | | **580** | **107** |

**Subset Construction.** Based on the above error analysis, computation pattern classification, and subject domain distribution, we constructed **SciBench-107**—a high-quality, balanced, and fair subset of 107 problems for evaluating our method.

*Table 7.* SciBench-107 subset balance across subject categories and computation patterns. The subset preserves diversity in both dimensions while concentrating on challenging cases.

| Subject | Direct Formula | ODE/ IVP | Root- Finding | Conserv. Laws | Numer. Methods | Other | Total |
|---|---|---|---|---|---|---|---|
| Classical Mech. | 8 | 5 | 3 | 3 | 1 | 1 | 21 |
| Diff. Equations | 4 | 7 | 2 | 0 | 2 | 1 | 16 |
| Thermodynamics | 6 | 0 | 1 | 1 | 1 | 1 | 10 |
| Statistics | 7 | 0 | 0 | 0 | 2 | 1 | 10 |
| Calculus | 5 | 0 | 1 | 0 | 3 | 1 | 10 |
| Quantum Mech. | 6 | 0 | 1 | 1 | 1 | 1 | 10 |
| Phys. Chemistry | 4 | 1 | 1 | 1 | 1 | 0 | 8 |
| Fund. Physics | 4 | 1 | 0 | 1 | 1 | 1 | 8 |
| Materials Sci. | 4 | 0 | 1 | 0 | 2 | 1 | 8 |
| Quantum Chem. | 4 | 0 | 0 | 0 | 1 | 1 | 6 |
| **Total** | **52** | **14** | **10** | **7** | **15** | **9** | **107** |
| **Percentage** | 48.6% | 13.1% | 9.3% | 6.5% | 14.0% | 8.4% | 100% |

Table 6 shows the subject distribution, and Table 7 demonstrates that our subset maintains balance across both subject domains and computation patterns.

# C. WebSocket Communication Protocol

This appendix provides technical details of the **Asynchronous State Synchronization Protocol** referenced in Section 3.3. The protocol enables the LLM agent to maintain continuous presence within the simulation environment by treating execution as a continuous time interval rather than a discrete function call.

## C.1. Protocol Overview

The WebSocket-based communication protocol implements three key capabilities that distinguish EmbodiedAct from traditional tool-calling agents:

1. **Asynchronous Acknowledgment**: Operations are confirmed immediately upon receipt, decoupling request transmission from execution completion.

2. **Streaming Observation**: The Operation Tracker pushes intermediate states $o_t$ via persistent WebSocket channel, enabling real-time monitoring during $t \in [t_{start}, t_{end}]$.

3. **Stateful Sessions**: Each session maintains operation history, pending verifications, and connection state across multiple interactions.

## C.2. Message Format Specification

All messages conform to the `EnhancedMessage` schema shown in Table 8.

*Table 8.* EnhancedMessage Schema Fields

| Field | Type | Description |
|---|---|---|
| id | string | Unique message identifier (UUID) |
| type | enum | Message type (see Table 9) |
| payload | object | Type-specific data payload |
| timestamp | float | Unix timestamp of message creation |
| session_id | string | Session identifier for routing |
| operation_id | string | Links related messages in a flow |
| status | enum | Current operation status |
| correlation_id | string | References parent message |

## C.3. Message Types

Table 9 enumerates all 16 message types organized by functional category.

*Table 9.* WebSocket Message Types

| Category | Message Type | Direction |
|---|---|---|
| *Operation Lifecycle* | `operation_request` | $\rightarrow$ |
| | `operation_ack` | $\leftarrow$ |
| | `operation_start` | $\leftarrow$ |
| | `operation_progress` | $\leftarrow$ |
| | `operation_complete` | $\leftarrow$ |
| | `operation_failed` | $\leftarrow$ |
| *Streaming Output* | `code_output` | $\leftarrow$ |
| | `code_status` | $\leftarrow$ |
| | `code_debug` | $\leftarrow$ |
| | `code_event` | $\leftarrow$ |
| *State Sync* | `model_state_update` | $\leftarrow$ |
| | `state_verification` | $\rightarrow$ |
| | `state_confirmed` | $\leftarrow$ |
| *Session Mgmt* | `session_init` | $\rightarrow$ |
| | `heartbeat` | $\leftrightarrow$ |
| | `error` | $\leftarrow$ |

$\rightarrow$: Agent to Server    $\leftarrow$: Server to Agent    $\leftrightarrow$: Bidirectional

## C.4. Streaming Monitoring Sequence

Figure 8 illustrates the complete cognitive loop during operation execution, showing how $\mathcal{M}_{code}$ dispatches actions and $\mathcal{M}_{perc}$ continuously monitors the observation stream to infer environment state $z_t$ before the final result is returned.

## C.5. Active Interruption with Hot-Fix Loop

Figure 9 demonstrates the active interruption capability. When $\mathcal{M}_{perc}$ detects an anomaly in the system events $\Psi_{sys}$ and infers $z_t = $ Warning, $\mathcal{M}_{code}$ generates an interrupt action $a_{stop}$ to halt execution. The Hot-Fix Loop then computes a repair action $a'_t \leftarrow \text{Repair}(a_t, o_t)$.

## C.6. Example Message Payloads

The following JSON snippets illustrate typical message payloads.

### 1. Operation Request (Agent $\rightarrow$ Server):

```
{"id": "a1b2c3d4-...", "type": "operation_request",
 "payload": {"operation_type": "execute_code",
  "parameters": {"script_name": "simulate_pid.m"}},
 "session_id": "sess-1234", "operation_id": "op-5678"}
```

### 2. Streaming Output (Server $\rightarrow$ Agent):

```
{"id": "msg-uuid", "type": "code_output",
 "payload": {"stdout": "Iteration 42: error=0.0023\n"},
 "operation_id": "op-5678", "status": "in_progress"}
```

### 3. Operation Complete (Server $\rightarrow$ Agent):

```
{"id": "msg-complete", "type": "operation_complete",
 "payload": {"result": {"success": true, "iterations": 156}},
 "operation_id": "op-5678", "status": "completed"}
```

## C.7. Operation Status State Machine

Figure 10 shows the state transitions for operation status tracking.

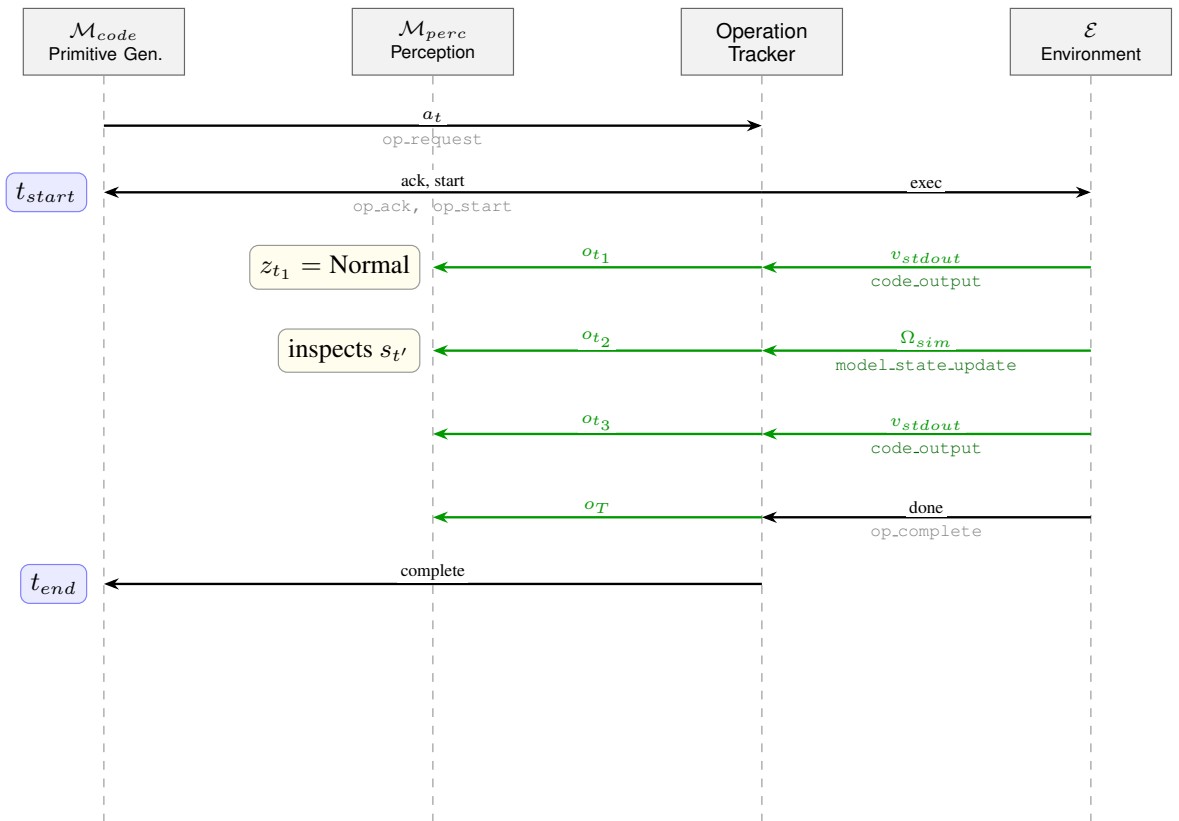

*Figure 8.* Streaming monitoring sequence showing the full cognitive loop. Message types (gray labels below arrows) correspond to Table 9. $\mathcal{M}_{code}$ generates action $a_t$, while $\mathcal{M}_{perc}$ continuously inspects the observation stream $o_t$ during $t \in [t_{start}, t_{end}]$ to infer intermediate states $s_{t'}$ and compute $z_t$.

## C.8. Design Considerations

The protocol addresses several challenges inherent to coupling LLM agents with long-running scientific simulations:

- **Timeout and Fault Tolerance**: Scientific simulations (e.g., PDE solvers, antenna optimization) may execute for extended periods. The protocol employs a configurable timeout (default: 600s) with automatic state transition to FAILED, enabling the agent to detect stalled operations and invoke recovery strategies.

- **Concurrent Operation Multiplexing**: The operation_id field implements a lightweight multiplexing scheme, allowing multiple operations to share a single WebSocket channel while maintaining message-operation affinity—a design choice that reduces connection overhead in multi-step reasoning chains.

- **Session Persistence**: To support iterative refinement workflows, the protocol maintains session state across transient disconnections. The client-side tracker preserves pending operations, enabling seamless resumption when connectivity is restored—critical for unreliable network environments.

- **Encoding Robustness**: Legacy simulation toolboxes may emit non-UTF-8 byte sequences. Rather than failing immediately, the protocol implements graceful degradation: after $k$ consecutive decoding failures (default $k=3$), the operation is marked complete with a warning, preserving partial results for downstream analysis.

- **Resource Management**: To prevent memory exhaustion during prolonged sessions, operation trackers are garbage-collected after $T_{timeout} + \Delta$ seconds ($\Delta=60$s by default), balancing resource efficiency against the need to retain context for late-arriving messages.

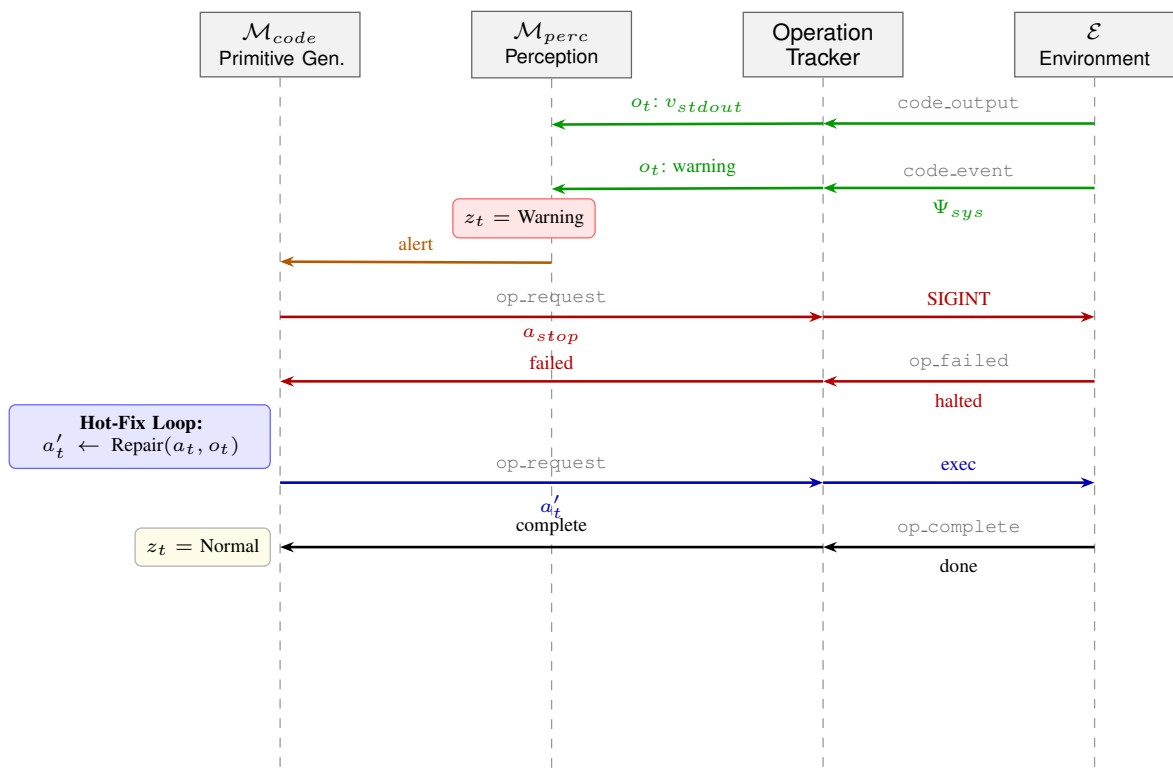

*Figure 9.* Active interruption with Hot-Fix Loop. When $\mathcal{M}_{perc}$ detects $z_t =$ Warning from system events $\Psi_{sys}$, it alerts $\mathcal{M}_{code}$, which generates interrupt action $a_{stop}$ (red arrows). The Hot-Fix Loop then computes repair action $a'_t \leftarrow \text{Repair}(a_t, o_t)$ (blue arrows) to recover.

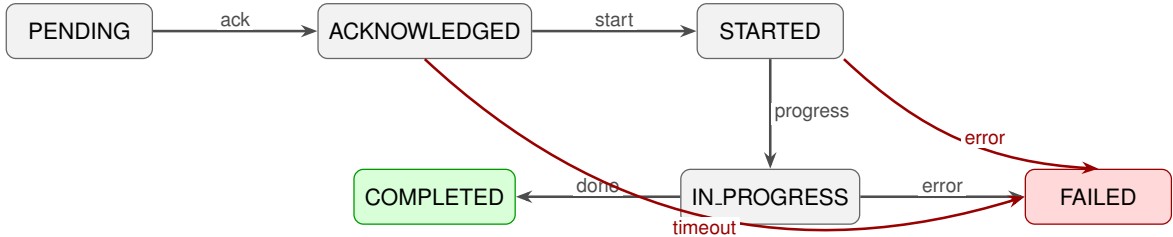

*Figure 10.* Operation status state machine. Normal flow: PENDING $\rightarrow$ ACKNOWLEDGED $\rightarrow$ STARTED $\rightarrow$ IN_PROGRESS $\rightarrow$ COMPLETED. Errors or timeouts transition to FAILED.

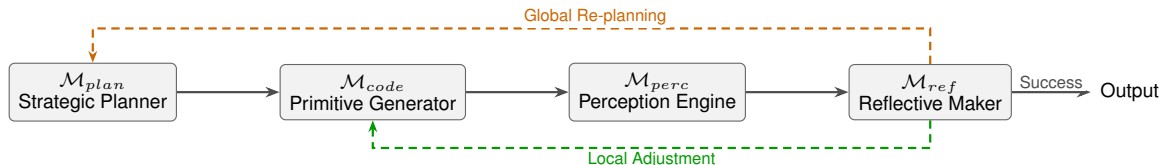

*Figure 11.* EmbodiedAct's cognitive loop with two feedback paths for self-correction.

## D. Case Study: Closed-Loop Self-Correction in EmbodiedAct

This appendix presents 9 representative cases demonstrating EmbodiedAct's three self-correction mechanisms (Figure 11), each illustrated with 2 successful cases and 1 limitation case.

### D.1. Runtime Perception and Error Recovery

**Core Value**: Detect and fix runtime errors (undefined functions, input/output mismatch) within the same problem episode, then continue solving.

---

**Case 1.1: Input/Output Mismatch Recovery**  ✓ Correct

**Problem.** *IVP:* $y' = ty(4 - y)/3$, $y(0) = 0.5$. *Find $T$ when $y(T) = 3.98$.*

**Trace.**
```
iter 0:  Error:  Number of inputs must match outputs.
iter 1:  T=3.29527274450143, yT=3.98, T_closed_form=3.29527274450382
         abs_diff_T=2.39e-12 ⇒ Result:  3.29527
```

**Outcome.** $T = 3.2953$ matches GT 3.29527.    **Mechanism.** $\mathcal{M}_{perc}$ detects function signature error → local adjustment fixes interface → self-verification via closed-form comparison.

---

**Case 1.2: Undefined Function Recovery**  ✓ Correct

**Problem.** *IVP:* $y' + \frac{1}{4}y = 3 + 2\cos 2t$, $y(0) = 0$. *Find $t$ when $y = 12$.*

**Trace.**
```
iter 0:  Error:  Function 'odefun' not recognized.
iter 1:  t_event=10.0657784374513, y_event=12 ⇒ Result:  10.0658
```

**Outcome.** $t = 10.0658$ matches GT 10.065778.    **Mechanism.** $\mathcal{M}_{perc}$ captures undefined function error → local adjustment defines missing function → no replan needed.

---

**Case 1.3: Limitation—Error Fixed but Semantic Extraction Wrong**  ✗ Wrong

**Problem.** *Elastic collision: find $u_1/u_2$ such that $m_1$ is at rest after collision.*

**Trace.**
```
iter 0-1:  Errors:  'nu1', 'nu2' undefined → fixed in iter 2
iter 2:  selected_r=2.414213562, alpha=0.414...  ⇒ Result:  2.4142
```

**Outcome.** Output $1 + \sqrt{2} \approx 2.414$ vs GT $(1 + \sqrt{2})^2 = 5.828$.    **Limitation.** Runtime recovery fixed execution errors but model extracted the wrong target quantity—multi-round error recovery *cannot* guarantee conceptual correctness.

---

### D.2. Local Iteration with Intermediate Variable Monitoring

**Core Value**: Monitor intermediate variables across iterations, pass information between rounds to solve problems without closed-form solutions—enabling parameter exploration and iterative refinement.

---

**Case 2.1: Iterative Threshold Verification (5% Criterion)**  ✓ Correct

**Problem.** *$Br_2$ vibrational wavenumber 323.2 $cm^{-1}$. At what $T$ is the partition function within 5% of the approximate formula?*

**Key Intermediates.**
```
theta_v=465.01, T_5pct=4493.795, q_exact=10.172, q_approx=9.664
rel_error_at_solution=0.05000 ⇒ Result:  4493.8
```

**Outcome.** $T = 4493.8$ (GT 4500, within 5%).    **Mechanism.** System explicitly computed and logged $q_{exact}$, $q_{approx}$, and $rel\_error = 0.05$—converting the "5% criterion" into a verifiable numerical check.

---

---

**Case 2.2: Dense Grid Search with Bracket Refinement**      ✓ Correct (Reflection)

**Problem.** *Spring-mass system: find $\tau$ after which $|u(t)| < 0.1$ for all $t > \tau$.*

**Initial Attempt (Wrong).** `tau=23.3134 (found` *a* `crossing, not the` *last* `one)`

**After Reflective Feedback (Correct).** `t_grid_count=200001, brackets_tL=25.6772, brackets_tR=25.6776`
`tau=25.6773, settling_ok=1 ⇒ Result: 25.6773 (GT exact)`

**Mechanism.** Dense grid search located last threshold crossing → bracket refinement → `settling_ok` post-hoc verification confirms $|u(t)| < 0.1$ for all $t > \tau$.

---

**Case 2.3: Limitation—Persistent Script Errors Despite Intermediate Passing**      ✗ Wrong

**Problem.** *Rocket trajectory with variable air density: determine max height.*

**Trace.**
```
iter 0:  total_mass, fuel_mass, mdot logged; result=8865.56 (unstable)
iter 1:  Error:  Script functions must end with 'end'
iter 2:  Error:  'node_options_burn' undefined
```

**Limitation.** Iteration 0 logged physical parameters, but subsequent iterations introduced MATLAB syntax errors. Intermediate monitoring cannot help if generated scripts are syntactically invalid—the model struggled to maintain coherence for this complex multi-stage problem.

## D.3. Answer Verification Feedback

**Core Value**: When $\mathcal{M}_{ref}$ (Reflective Decision Maker) judges that the output answer may be incorrect—despite no runtime errors—it automatically triggers a reflective re-attempt with diagnostic feedback injected into context. This mechanism addresses "no error but wrong answer" samples by prompting the model to revise its strategy or correct modeling assumptions, *without manual intervention*.

---

**Case 3.1: Sanity Check Triggered Replan**      ✓ Correct

**Problem.** *Model rocket: speed at burnout is 131 m/s. How far has it traveled?*

**Key Intermediates.** `v_burnout_calc=133.56, x_burnout=112.24, velocity_error_pct=1.95%`
`replan_count=1, error_types=[ReviewFailed] ⇒ Result: 112.24 m`

**Outcome.** $x = 112.24$ m (GT 108, within 5%).      **Mechanism.** $\mathcal{M}_{ref}$ triggered `ReviewFailed` → replan → model added sanity check comparing computed $v_{burnout}$ with given value to increase confidence.

---

**Case 3.2: Concept Error Corrected via Reflective Judgment**      ✓ Correct (Reflection)

**Problem.** *Clown juggles 4 balls, 0.9 s/cycle. What is minimum throw speed?*

**Initial Attempt (Wrong).** `T_flight=3.6 (used` $N \cdot t_{cycle}$`), v0=17.658 (GT: 13.2)`

**After Reflective Feedback (Correct).** `T_flight=2.7 (used` $(N-1) \cdot t_{cycle}$`), v0=13.2435`

**Diagnostic Context Injected by $\mathcal{M}_{ref}$.** `Previous Answer: 17.658 flagged as suspicious. Hint: internally consistent but likely wrong.`

**Mechanism.** $\mathcal{M}_{ref}$ judged the initial answer suspicious and automatically triggered re-attempt with diagnostic feedback → model reconsidered modeling assumption (off-by-one: ball must stay airborne for $(N-1)$ cycles) → corrected to $v_0 = g \cdot T/2 = 13.24$ m/s.

---

**Case 3.3: Limitation—Systematic Assumption Mismatch**      ✗ Wrong

**Problem.** *Barometric formula: pressure at 11 km altitude.*

**Initial & Reflection-triggered Attempts.** `Both:  p=0.271 atm (isothermal barometric formula) vs GT=0.72 atm`

**Limitation.** Both attempts used the isothermal model $p = p_0 \exp(-Mgh/RT)$ with standard parameters, yielding consistent $\sim$0.27 atm. The GT implies a different model (non-isothermal or different parameters). When conceptual approach is fundamentally misaligned with ground truth's unstated assumptions, even reflective re-attempts cannot help—this is a benchmark annotation ambiguity, not a framework failure.

## D.4. Summary

*Table 10.* Summary of 9 case studies demonstrating EmbodiedAct's self-correction mechanisms.

| Category | Case | Mechanism | Outcome |
|---|---|---|---|
| 1. Runtime Error Recovery | 1.1 | I/O mismatch → Local Adjustment | ✓ |
| | 1.2 | Undefined function → Local Adjustment | ✓ |
| | 1.3 | Errors fixed, semantic extraction wrong | ✗ |
| 2. Intermediate Monitoring | 2.1 | 5% threshold verification | ✓ |
| | 2.2 | Grid search + reflective refinement | ✓ |
| | 2.3 | Script structure errors persisted | ✗ |
| 3. Answer Verification | 3.1 | Sanity check → Replan | ✓ |
| | 3.2 | $\mathcal{M}_{ref}$ judges → Reflective Re-attempt | ✓ |
| | 3.3 | Systematic assumption mismatch | ✗ |

**Key Takeaways:**

- **Runtime Perception** enables fixing execution errors within the same episode—but cannot catch semantic/conceptual errors.

- **Intermediate Monitoring** supports iterative solving for problems without closed-form solutions—but requires syntactically valid scripts.

- **Answer Verification Feedback** enables $\mathcal{M}_{ref}$ to automatically detect "no error but suspicious answer" cases and trigger reflective re-attempts that prompt strategy revision—all within the framework's closed loop, without manual intervention. However, it cannot overcome fundamental assumption mismatches with ground truth.

# E. Computational Resource Analysis

This appendix provides a detailed analysis of token consumption across different methods, offering transparency about the computational overhead of the embodied cognitive architecture.

## E.1. Token Usage Comparison

Table 11 compares token consumption across three paradigms on multi-step reasoning tasks requiring iterative refinement.

*Table 11.* Token usage comparison on multi-step reasoning tasks. All ratios are relative to Baseline.

| Method | Total Tokens | Ratio | Avg Score | Pass Rate |
|---|---|---|---|---|
| Baseline (Direct) | 104,889 | 1.0× | 53.3 | 33.3% |
| CodeAct | 657,141 | 6.3× | 44.0 | 15.6% |
| **EmbodiedAct** | **1,115,740** | **10.6×** | **61.6** | **51.1%** |

**Key Observations.**

- **More tokens $\neq$ effective iteration**: CodeAct consumes 6.3× more tokens than Baseline but achieves *lower* accuracy (44.0 vs. 53.3). For multi-step reasoning tasks, CodeAct's brief action-observation cycles lack sufficient context to form effective iterative refinement.

- **Justified overhead**: EmbodiedAct's 10.6× overhead relative to Baseline yields +17.6 points in average score and +35.5 percentage points in pass rate over CodeAct.

## E.2. Token Distribution by Module

Table 12 breaks down EmbodiedAct's token consumption across its four cognitive modules, revealing the computational allocation of the neuro-functional architecture.

**Module-Level Insights.**

*Table 12.* Token distribution by module. $\mathcal{M}_{plan}$ and $\mathcal{M}_{ref}$ show similar consumption due to symmetric input contexts and comparable output schemas.

| Module | Symbol | Input | Output | Total | % |
|---|---|---|---|---|---|
| Strategic Planner | $\mathcal{M}_{plan}$ | 124,764 | 9,264 | 134,028 | 12% |
| Primitive Generator | $\mathcal{M}_{code}$ | 601,858 | 44,778 | 646,636 | 58% |
| Runtime Perception | $\mathcal{M}_{perc}$ | 187,146 | 13,894 | 201,040 | 18% |
| Reflective Decision | $\mathcal{M}_{ref}$ | 123,772 | 10,264 | 134,036 | 12% |
| **Total** | – | 1,037,540 | 78,200 | 1,115,740 | 100% |

- $\mathcal{M}_{code}$ **dominance (58%)**: The majority of tokens are consumed by the Primitive Generator, reflecting multi-iteration code synthesis and refinement cycles typical in engineering design tasks.

- **Balanced perception-reflection (30%)**: The combined overhead of $\mathcal{M}_{perc}$ (18%) and $\mathcal{M}_{ref}$ (12%) enables runtime state monitoring and adaptive replanning.

- **Efficient planning (12%)**: $\mathcal{M}_{plan}$ consumes only 12% of tokens, as structured recipe generation front-loads reasoning into a compact JSON schema.

**Verifiability-Efficiency Trade-off.** The token overhead represents a deliberate architectural choice: EmbodiedAct maintains rich execution context (intermediate states, error traces, constraint violations) to detect failures early and correct course mid-execution. This investment yields higher accuracy (61.6% vs. 44.0% for CodeAct) and better reliability (51.1% pass rate vs. 15.6%), demonstrating that *intelligent* token usage—with streaming perception and modular cognition— outperforms *blind* multi-turn iteration.

