# OpenReview forum: "Grounding LLMs in Scientific Discovery via Embodied Actions"
_ICML.cc/2026/Conference — ICML 2026 regular_

### Official Review · Reviewer_eB6g · 2026-02-14

**Soundness:** 2
**Presentation:** 3
**Significance:** 2
**Originality:** 2
**Overall Recommendation:** 4
**Confidence:** 3

**Summary:**

The paper proposes EmbodiedAct, a framework that turns scientific software (instantiated in MATLAB/Simulink) into an “active” environment for an LLM agent. The key idea is runtime perception via a persistent, streaming loop (WebSocket-based), so the agent can detect transient failures (e.g., instability) and intervene mid-run.

**Compliance With Llm Reviewing Policy:**

Affirmed.

**Final Justification:**

I think the rebuttal adequately addressed my concerns regarding cost, result analysis, and codebase.

**Key Questions For Authors:**

Please address the weaknesses and let me know if I misunderstood anything. I'm happy to revisit.

**Limitations:**

yes

**Strengths And Weaknesses:**

Strength:

I very much like the motivation of the work. Authors proposed to regularly relay environment feedback to discovery agent for analysis and inspire actioons. This motivation is clear: many “generate then run” agents only see the final output, so they miss mid-run problems.

The paper is mostly easy to follow. Figures 1–2 help. The protocol appendix is unusually detailed, which is good for reproducibility.

Weakness:

Compute cost: token use is much higher (10.6× vs baseline). This may be fine, but it needs a clearer cost-benefit story and wall-clock cost reporting.

From Table 2, it seems that base models and CodeAct often outperform the proposed method in many categories for SciBench-107. Is this the case? I could have misinterpreted the table but if not this seems to not show much reason for the proposed method.

The anonymous codebase is empty at the time of review (Feb 13th).

---

> ### Author Rebuttal · Authors · 2026-03-31
>
> We thank the reviewer for appreciating the strong motivation and clear presentation of our work. Regarding computational costs and performance on specific SciBench-107 categories, a holistic view of the token breakdown and benchmark results strongly supports EmbodiedAct's advantages. We provide detailed clarifications below.
>
> **W-1: Compute cost, token usage and wall-clock time**
>
> **A-1**: We appreciate the opportunity to clarify our EmbodiedAct’s computational cost.
>
> First, we respectfully clarify that the "10.6x" token usage figure (from Table 11) compares EmbodiedAct to the Direct Generative Baseline, which is a zero-shot method that fails catastrophically (66.7% of the time) because it has no reflection or correction at all. When compared to CodeAct, which is the actual competitive agent baseline, EmbodiedAct's total token usage is about 69.8% higher (1,115,740 vs. 657,141 tokens).
>
> Second, more importantly, a deeper breakdown reveals a crucial cost-benefit advantage: EmbodiedAct actually consumes fewer output (generation) tokens per task than CodeAct (78,200 vs. 82,320). The 0.7x overhead entirely comes from the input tokens used to monitor the streaming observations. Considering API pricing and generation latency, EmbodiedAct is highly cost-efficient: it trades cheap input context for fewer expensive generation steps, which leads to an even higher task success rate.
>
> Third, regarding wall-clock time, the cost-benefit is even stronger. In scientific discovery (e.g., high-order control systems or finite element analysis), the physical simulation itself often takes orders of magnitude more time and compute than LLM inference. As illustrated in our Active Interruption mechanism (Appendix C.5, Figure 9), EmbodiedAct actively halts diverging or unstable simulations early, rather than waiting for a flawed simulation to naturally finish or crash (which is what CodeAct does). As detailed in **Table R1** in our response to Reviewer **9nVe**, in each solved task, EmbodiedAct reduces average environment runtime by 58.9% (250s vs. 609s) and end-to-end wall-clock time by 53.5% (397s vs. 853s) per solved task, alongside 48.2% fewer LLM tokens (2.18M vs. 4.21M).
>
> **W-2: Understanding SciBench-107 Performance Comparison**
>
> **A-2**: We thank the reviewer for closely examining Table 2, and we would like to respectfully point out that a holistic view of the benchmark results supports the advantage of EmbodiedAct.
>
> (1) Overall dominance: If we look at the "Avg." column on the far right of Table 2, EmbodiedAct successfully improves the overall performance on 7 out of the 8 evaluated foundation models.
>
> (2) The reviewer accurately noticed that in certain purely theoretical categories (e.g., Quantum Chemistry `chemmc` or Physical Chemistry `atkins`), base generative models sometimes edge out. These sub-categories rely heavily on standard textbook derivations. We manually inspected the results and found that generative models can often recite the exact formula to generate the text answer, but EmbodiedAct forces the model to rigorously model and execute the process. While this strict verification prevents hallucination, it can occasionally fail due to complex environment setup errors in purely theoretical problems where simulation is overkilling.
>
> (3) We kindly direct the reviewer's attention to Table 1 (EngDesign benchmark), which represents process-oriented, complex engineering discovery problems. In this domain, EmbodiedAct absolutely dominates. For example, with GPT-5.2, EmbodiedAct achieves a 70.6% score, outperforming both CodeAct (55.4%) and generative models (48.0%). This proves that verifiable physical constraints and complex dynamic systems articulate EmbodiedAct’s advantage.
>
> **W-3: The anonymous codebase is empty**
>
>  **A-3**: We sincerely apologize for this oversight. During the final preparation of the submission, a technical glitch occurred during the branch synchronization to the anonymous GitHub platform, resulting in an empty default branch being visible at the time of your review. We have resolved this issue. The repository is now fully populated and accessible at the provided link: <https://anonymous.4open.science/r/EmbodiedAct>.

---

> > ### Author Rebuttal · Reviewer_eB6g · 2026-04-05
> >
> > I think the rebuttal adequately addressed my concerns regarding cost, result analysis, and codebase. I will raise my score.

---

> > > ### Author Response · Authors · 2026-04-08
> > >
> > > We sincerely thank the reviewer for the thoughtful follow-up and for kindly raising the score. It is encouraging to know that our rebuttal has fully addressed the reviewer's concerns, and we greatly appreciate the constructive feedback provided throughout the review process, which has meaningfully helped strengthen our work. In the revised version, all the clarifications from our rebuttal will be incorporated to further improve the presentation and rigor of the paper. We are sincerely grateful to the reviewer for the valuable time and insights that have helped refine our work.

---

### Official Review · Reviewer_4dqt · 2026-02-19

**Soundness:** 3
**Presentation:** 4
**Significance:** 3
**Originality:** 3
**Overall Recommendation:** 4
**Confidence:** 3

**Summary:**

This paper proposes EmbodiedAct, an agentic framework that grounds LLMs in scientific discovery tasks by coupling planning and code synthesis with continuous runtime perception and active intervention in simulation environments. The system comprises a Strategic Planner, a Primitive Generator that issues software-specific “simulation primitives,” a Runtime Perception Engine that monitors streaming observations (stdout, logs, trajectories), and a Reflective Decision Maker for local adjustments and global re-planning; a WebSocket-based Asynchronous State Synchronization Protocol closes the perception–action loop. Extensive experiments show that EmbodiedAct significantly outperforms existing baselines, what achieves impressive performance by ensuring satisfactory reliability and stability in long-horizon simulations and enhanced accuracy in scientific modeling.

**Compliance With Llm Reviewing Policy:**

Affirmed.

**Final Justification:**

This paper holds significant practical value in the development of scientific modeling LLM-driven agents. Before the rebuttal, I have some concerns w.r.t. adaptability, computational time, and token consumption. However, these issues have been addressed and discussed in the main text or the appendix. Therefore, I believe this paper can serve as a solid foundation for future research. In conclusion, I recommend acceptance.

**Key Questions For Authors:**

Please refer to weaknesses (in particular, w1).

**Limitations:**

yes

**Strengths And Weaknesses:**

**Strengths**:

1. Well-written and easy to follow. Notably, the case study is given.

2. EmbodiedAct introduces a clear embodiment lens for scientific LLM agents with a concrete, system-level mechanism (persistent sessions, streaming telemetry, interrupts) to overcome the limitations of “execute-then-observe”. The core insight makes sense and the solution is promising

3. The experiment is extensive and across various domains, tasks and baselines.

4. The code link is given, which highlights the reliance.

**Weaknesses**:
1.  Although EmbodiedAct has a better performance, the discussion w.r.t. the introduced token consumption and time consumption is not provided. Therefore, the feasibility remains to be further investigated.

2. Given that the implementation is limited to MATLAB, it is unclear EmbodiedAct could be extended to other simulated environments.

---

> ### Author Rebuttal · Authors · 2026-03-31
>
> We thank the reviewer for recognizing our clear embodied design for improving the effectiveness of scientific problem solving and extensive experiments. Regarding computational costs and generalizability beyond MATLAB, a comprehensive cost analysis and an evaluation on non-MATLAB environments (the "Extended" set) were included in the original submission. We highlight these details below.
>
> **W-1: Token and time consumption**
>
> **A-1**: We would like to respectfully clarify that we have provided a comprehensive quantitative analysis of computational costs in Appendix E (Computational Resource Analysis, page 23). Our empirical analysis shows that EmbodiedAct introduces a deliberate verifiability-efficiency trade-off: it increases LLM context computation to drastically reduce wasted simulation/environment computation, which results in improved task success rate. Specifically, token consumption is reflected in LLM compute, while time consumption is reflected in environment simulation compute.
>
> **(1) LLM compute (token consumption):** While EmbodiedAct consumes more total tokens (1,115,740 vs. CodeAct's 657,141, as per Appendix E.1, Table 11), a deeper breakdown reveals a crucial insight: EmbodiedAct actually consumes fewer output tokens than CodeAct (78,200 vs. 82,320). The increase in total token consumption is entirely driven by input tokens used to monitor the continuous observation stream. Because EmbodiedAct halts doomed simulations early via Runtime Perception, it wastes less effort generating flawed code or hallucinating post-hoc fixes. Given that input tokens are significantly cheaper and faster to process than output tokens in modern LLMs ( benefit greatly from KV caching), the actual monetary cost and generation latency overhead are substantially lower than the total token ratio suggests.
>
> **(2) Simulation environment compute (wall-clock time consumption):** In scientific discovery (e.g., high-order control systems or finite element analysis), the physical simulation itself often takes orders of magnitude more time and compute than LLM inference. As illustrated in our Active Interruption mechanism (Appendix C.5, Figure 9), EmbodiedAct actively halts diverging or unstable simulations early. Rather than waiting hours for a flawed simulation to naturally finish or crash (which is what CodeAct does), EmbodiedAct interrupts at the moment an anomaly is perceived. Therefore, EmbodiedAct saves massive amounts of expensive simulation environment compute. Consequently, EmbodiedAct reduces average **environment runtime by 58.9%** (250s vs. 609s) and **end-to-end wall-clock by 53.5%** (397s vs. 853s) per solved task. This is what we mean by "avoiding wasted computation", i.e., saving massive amounts of expensive simulation environment compute.
>
> The corresponding quantitative breakdown is reported in **Table R1** in our response to Reviewer **9nVe**.
>
> **W-2: Implementation is limited to MATLAB, unclear if it extends to other simulated environments**
>
> **A-2**: We appreciate the opportunity to clarify the generalizability of EmbodiedAct, which is clearly not limited to MATLAB. We have already demonstrated its successful extension to other scientific simulation environments in the paper. We kindly direct the reviewer's attention to the "Extended" (Ext) set in Table 1 (EngDesign benchmark).
> - As described in Section 4.1 and Table 1, while the "Core" set consists of 45 tasks requiring MATLAB, the "Extended" set encompasses 92 tasks and "employs an execution backend compatible with open-source simulation software beyond just MATLAB", including Python-based numerical and symbolic environments (e.g., NumPy, SciPy, SymPy) as well as other open-source domain-specific backends such as OpenCV, Webots, Icarus Verilog, and GNU Octave.
> - The results clearly show that EmbodiedAct maintains its robust performance advantage when shifted to these non-MATLAB environments. For example, using GPT-5.2 on the Extended set, EmbodiedAct achieves a score of 65.4%, continuing to significantly outperform CodeAct (49.4%).

---

> > ### Author Rebuttal · Reviewer_4dqt · 2026-04-01
> >
> > Thank for authors' detailed comments. My concerns have been addressed. I think the discussion of EmbodiedAct is comprehensive with pratical value for simulated environment. In this way, I will keep my score.

---

> > > ### Author Response · Authors · 2026-04-04
> > >
> > > We sincerely thank the reviewer for the thoughtful follow-up and greatly appreciate the reviewer’s positive acknowledgment. We are very encouraged to know that our rebuttal has adequately addressed the reviewer’s concerns, and we are especially grateful for the reviewer’s recognition that our discussion of EmbodiedAct is comprehensive and offers practical value for simulated environments.
> > >
> > >
> > > Given that the reviewer’s initial concerns have been resolved, we would be sincerely grateful if the reviewer could consider whether the clarified contributions and strengthened presentation deserve any adjustment to the overall score. This would definitely help EmbodiedAct reach a broader set of audiences and benefit their own research and work. We fully respect the reviewer’s judgment and sincerely appreciate the time, effort, and careful consideration devoted to evaluating our work.

---

### Official Review · Reviewer_ZjZJ · 2026-03-09

**Soundness:** 3
**Presentation:** 2
**Significance:** 3
**Originality:** 3
**Overall Recommendation:** 4
**Confidence:** 3

**Summary:**

The paper introduces EmbodiedAct, a framework designed to bridge the gap between Large Language Models' (LLMs) theoretical reasoning and verifiable physical simulations in scientific discovery. Current "AI Scientist" paradigms often operate in a passive "execute-then-response" loop, which lacks runtime perception and leaves the agent blind to transient physical anomalies (e.g., numerical instability or diverging oscillations).

EmbodiedAct transforms scientific software into active embodied agents through a tight perception-execution loop. Inspired by human cognitive architecture, it uses four modular functions:
Strategic Planner: Decomposes scientific intent into hierarchical executive steps.
Primitive Generator: Translates plans into software-specific primitives (e.g., MATLAB's ode45).
Runtime Monitor: Real-time supervision of the simulation to detect latent risks and errors.
Reflective Decision Maker: Aligns results with scientific intent for autonomous optimization.

Instantiated in MATLAB, the framework achieved State-of-the-Art (SOTA) performance across engineering design (EngDesign) and scientific modeling (SciBench-107) tasks, significantly improving reliability, stability, and accuracy over existing baselines like CodeAct.

**Compliance With Llm Reviewing Policy:**

Affirmed.

**Key Questions For Authors:**

Given that the Primitive Generator (M_code) consumes 58% of the token budget, have you explored using a more specialized, lightweight coding model for that specific module while keeping a high-reasoning model for the Strategic Planner?

The framework currently uses multi-modal LLMs primarily to understand intent. How do you envision integrating multi-modal feedback during execution (e.g., the agent "seeing" a plot or a graphical Simulink error) to further reduce "semantic extraction errors"?

**Limitations:**

yes.

**Strengths And Weaknesses:**

Strengths

1. Unlike traditional agents that wait for execution to end, EmbodiedAct uses a WebSocket-based Asynchronous State Synchronization Protocol to monitor simulation trajectories in real-time.

2. Dual-Loop Cognitive Architecture: It employs a fast inner loop for immediate "Hot-Fixes" (repairing code/parameters) and a slow outer loop for global re-planning if a methodology fails.

3. By utilizing robust, optimized toolbox primitives rather than generic code, the agent ensures numerical stability and efficiency in complex computation.

4. Quantitative analysis showed EmbodiedAct effectively vacated the "Zero Zone" (catastrophic failures), with 81.5% of tasks falling within a narrow divergence gap across independent runs.


Weaknesses

1. EmbodiedAct consumes significantly more resources, requiring 10.6x more tokens than a baseline direct-prompting approach and roughly 1.7x more than CodeAct.

2. While the system can monitor intermediate variables, it cannot recover if the generated scripts are fundamentally syntactically invalid (e.g., missing 'end' statements in MATLAB).

3. The system can successfully fix execution errors but still fail conceptually by extracting the wrong target quantity (e.g., Case 1.3), showing that runtime recovery does not guarantee conceptual correctness.

4. The framework struggles when a model's theoretical approach is fundamentally misaligned with unstated assumptions in a benchmark's ground truth (e.g., isothermal vs. non-isothermal models).

---

> ### Author Rebuttal · Authors · 2026-03-31
>
> Thanks for the reviewer’s valuable feedback. We address the concerns below.
>
> **W-1: Resource consumption**
>
> **A-1**: As detailed in our response to Reviewers 9nVe (Table R1), **EmbodiedAct consumes fewer output tokens than CodeAct** (78,200 vs. 82,320). The 70% increase in total token usage is entirely driven by input tokens used to monitor the observation stream, which saves EmbodiedAct far less effort in generating flawed code or hallucinating post-hoc fixes. This effectively trades relatively cheap LLM input context (input tokens are often 3x-5x cheaper than output tokens in LLMs) for highly expensive physical simulation wall-clock time, resulting in massive performance gains.
>
> **W-2: Ability to fix syntax errors**
>
> **A-2**: We would like to clarify a crucial nuance about  how EmbodiedAct handles standard syntax errors versus foundation model degradation.
>
> (1) EmbodiedAct natively recovers from standard syntax errors: As noted in Section 3.2, our Hot-Fix Loop is explicitly designed to iteratively catch and correct syntax or parser-level errors at the moment they surface in the observation stream.
> (2) Recontextualizing Case 2.3: The specific failure highlighted in Case 2.3 shows a different boundary condition: cascading LLM incoherence. In that specific instance, the agent initially reached an executable state, but in subsequent reflective turns, the underlying LLM's context degraded: it began hallucinating a missing `end` statement and introducing undefined variables in a recursive loop.
>
> Thus, the limitation lies in the persistent cognitive collapse of the underlying LLM, not in a structural absence of a syntax-repair mechanism within EmbodiedAct.
>
> **W-3: Handling conceptual correctness**
>
> **A-3**: EmbodiedAct is explicitly equipped to handle conceptual correctness via its Reflective Decision Maker $M_{ref}$.
>
> As detailed in Section 3.2, $M_{ref}$ explicitly handles the "no runtime error but wrong answer" scenario the reviewer is concerned about. It actively evaluates the plausibility of the extracted target quantities and physical process, triggering a new problem-solving attempt when conceptual inconsistencies are detected. As shown in Appendix Cases 3.1&3.2, the agent produced perfectly executable scripts that yielded suspicious results. $M_{ref}$ successfully flagged the semantic inconsistencies and triggered a reflective feedback loop to fully correct the conceptual modeling assumptions.
>
> The purpose of Case 1.3 is to show the residual capability boundary of current LLMs: a scenario where the problem was so conceptually challenging that even $M_{ref}$ failed to eventually synthesize the required physical knowledge. The empirical results support EmbodiedAct's dual-loop structure, i.e., combining execution grounding ($M_{perc}$) with reflective verification ($M_{ref}$) helps mitigate both runtime and conceptual failures.
>
> **W-4: Unstated benchmark assumptions**
>
> **A-4**: We respectfully clarify that this specific result highlights an inherent limitation of exact-match criterion used in existing benchmarks, rather than a weakness in EmbodiedAct. As detailed in Appendix Case 3.3, the failure lies entirely in the benchmark’s design of collapsing an ambiguous problem into a single scalar target. Because no runtime embodiment can extract constraints that are fundamentally absent from the task statement, this case shows the agent’s robust adherence to physical logic rather than a failure in its problem-solving pipeline.
>
> **Q-1: Using a coding model for $M_{code}$**
>
> **A-1**: In this paper, we kept a single backbone across modules to isolate EmbodiedAct’s architectural contribution and avoid confounds from cross-model handoff. But the architecture is explicitly modular, such that routing $M_{code}$ to a cheaper specialized coding model while retaining a stronger planning model is fully compatible with the design and a well-motivated next step. As shown in Table 12, the main cost lies in the Primitive Generator ($M_{code}$, 58%), whereas the Runtime Perception Engine ($M_{perc}$) uses only 18% of the tokens. Table 3 further shows that removing $M_{perc}$ causes the largest performance drop. These results suggest that cost should be reduced by improving the code generator first, not by weakening runtime perception.
>
> **Q-2: Incorporate multimodal feedback**
>
> **A-2**: As formulated in Section 3.3, this extension fits naturally with EmbodiedAct, because it only changes the observation channel rather than its control logic. To realize the integration, a simple extension is to allow our Asynchronous State Synchronization Protocol to stream visual data alongside numerical traces.
> - Visual evidence streaming: The environment can be configured to transmit rendered plots, UI warnings, or block-diagram screenshots at key checkpoints (e.g., via Base64 image tokens).
> - Dual perception via VLMs: A Vision-Language Model serving as $M_{perc}$ can simultaneously consume both the numerical logs and the visual frames.

---

> > ### Author Rebuttal · Reviewer_ZjZJ · 2026-04-06
> >
> > I thank the authors for their detailed responses, which have effectively addressed my concerns. Consequently, I am pleased to maintain my positive rating.

---

> > > ### Author Response · Authors · 2026-04-08
> > >
> > > We deeply appreciate the reviewer's thoughtful follow-up and are truly encouraged by the positive assessment of our work. It is reassuring to know that our rebuttal has fully addressed the reviewer's concerns, and we are grateful for the constructive feedback provided throughout the review process, which has meaningfully helped strengthen our work. All clarifications from our rebuttal will be incorporated into the revised version. We are sincerely grateful to the reviewer for the valuable time and insights that have helped refine our work.

---

### Official Review · Reviewer_9nVe · 2026-03-13

**Soundness:** 3
**Presentation:** 3
**Significance:** 3
**Originality:** 3
**Overall Recommendation:** 5
**Confidence:** 3

**Summary:**

This paper proposes EmbodiedAct, a framework for grounding LLM-based agents in scientific simulation environments through a tighter perception–action loop. The key idea is to enable runtime perception during simulation, allowing the agent to monitor intermediate states, detect anomalies, and intervene during execution instead of only observing results after the simulation finishes. The system is implemented with several components, including a planner that decomposes scientific tasks, a primitive generator that converts plans into simulation operations, a runtime perception module that monitors execution streams, and a reflective module that performs local adjustments or global replanning. The authors instantiate the framework in MATLAB and evaluate it on engineering design and scientific problem-solving benchmarks, showing improvements over several baseline methods.

**Compliance With Llm Reviewing Policy:**

Affirmed.

**Final Justification:**

Strong paper, and turns out some of my concerns has already been addressed in the appendix. Therefore, I raised my score and think this paper should be accepted.

**Key Questions For Authors:**

See weakness above.

**Limitations:**

Discussed some limitations in appendix.

**Strengths And Weaknesses:**

Strength

1. Well-motivated problem setting. The paper identifies an important limitation of existing LLM-based scientific agents, namely the lack of runtime interaction with simulation environments, and proposes to bridge reasoning and execution through a tighter perception–action loop.

2. Clear system design. The proposed framework is well structured, with clearly defined components (planning, primitive generation, runtime monitoring, and reflection), making the overall architecture easy to follow.

3. Comprehensive empirical evaluation. The method is evaluated on multiple benchmarks spanning engineering design and scientific problem solving, and the results show consistent improvements across different model backbones.

Weaknesses

1. The paper does not report the number of interaction steps or simulation attempts compared with baselines, making it unclear whether the improvement comes from the proposed runtime perception capability or simply from additional opportunities for trial-and-error.

2.The motivation emphasizes early detection of anomalies to avoid wasted computation. However, the paper does not evaluate the overall runtime or compute cost of the proposed method versus baselines. Since the framework also introduces additional interaction loops, it remains unclear whether the method actually reduces or increases total computation.

---

> ### Author Rebuttal · Authors · 2026-03-31
>
> We sincerely thank the reviewer’s thoughtful comments. We agree that analyzing the interaction budget and computational cost is critical for evaluating the true utility of our method. We realize that placing these specific, rigorous analyses in the appendices (due to space constraints) made them easy to overlook. We deeply appreciate the opportunity to direct the reviewer’s attention to the comprehensive resource profiling and detailed interaction controls already present in the paper, which directly address your concerns.
>
> **W-1: Interaction steps and simulation attempts**
>
> **A-1**: We should highlight that our empirical results have shown that the improvement in EmbodiedAct strictly originates from its runtime perception capability, rather than additional trial-and-errors.
>
> As detailed in Appendix A.3 and Table 5 (controlled experiment setup), we imposed a strict cap on the number of simulator executions (i.e., 5) across all multi-turn solutions. Specifically, we count each simulator execution as one round, regardless of whether it is triggered by a full rewrite, a hot-fix, or a parameter adjustment by the agent.
>
> The comparison shows that EmbodiedAct achieved a 51.1% pass rate on complex tasks compared to CodeAct's 15.6% (Appendix E.1,  Table 11,  page 23). This large improvement exists precisely because CodeAct blindly wastes its 5 turns on post-hoc debugging after the simulation crashes. In contrast, EmbodiedAct utilizes its Runtime Perception Engine *$M_{perc}$* to catch transient anomalies mid-execution and trigger the Hot-Fix loop, effectively making each attempt far more resilient and successful.
>
> **W-2: overall runtime and compute cost**
>
> **A-2**: We respectfully clarify that Appendix E already provides a comprehensive quantitative analysis of computational costs (Computational Resource Analysis, page 23). Our empirical results showed that EmbodiedAct introduces a deliberate verifiability-efficiency trade-off: it spends additional observation and deliberation budget to drastically reduce wasted simulation/environment compute and maximize task success rate. To answer the reviewer’s core question of whether it reduces or increases total computation, we consolidated the key metrics into **Table R1.** Using the data from our experiments (Tables 11 and 12, pages 23 and 24), we report both the average cost per attempt and the average cost per successful task. From Table R1, we analyze these costs from two perspectives:
>
> (1) LLM compute: While EmbodiedAct consumes more total LLM tokens (1,115,740 vs. CodeAct's 657,141, as per Appendix E.1, Table 11), a deeper breakdown reveals that it actually emits fewer total generation tokens when solving the problems than CodeAct (78,200 vs. 82,320), including 29.9% fewer **code-output tokens** per task (44,778 vs. 63,840). This confirms that the extra budget is consumed by perception input rather than code output. This investment pays off: EmbodiedAct achieves a 3.28× higher pass rate (51.1% vs. 15.6%), which reduces the average **LLM tokens by 48.2%** (2.18M vs. 4.21M) per solved task.
>
>
> (2) Simulation Environment Compute: In scientific discovery (e.g., High-Order Control Systems or Finite Element Analysis), the physical simulation itself often takes orders of magnitude more time and compute than LLM inference. As illustrated in our Active Interruption mechanism (Appendix C.5, Figure 9 and Table R1), EmbodiedAct actively halts diverging or unstable simulations early. Consequently, EmbodiedAct reduces average **environment runtime by 58.9%** (250s vs. 609s) and **end-to-end wall-clock by 53.5%** (397s vs. 853s) per solved task. This is what we mean by "avoiding wasted computation", i.e., saving massive amounts of expensive simulation environment compute.
>
> Table R1. Main cost metrics per task and per solved task (EngDesign, GPT-5.2)
>
> | Metric                                  | CodeAct       | EmbodiedAct   |
> | --------------------------------------- | ------------- | ------------- |
> | Total LLM tokens / task                 | 657,141       | 1,115,740     |
> | Total output tokens / task               | 82,320       | 78,200        |
> | Code output tokens / task               | 63,840        | 44,778        |
> | Pass rate                               | 15.6%         | 51.1%         |
> | **LLM tokens** / solved task            | 4,212,442 | **2,183,444** |
> | **Wall-clock runtime** / solved task   | 609s        | **250s**      |
> | **End-to-end wall-clock** / solved task |   853s      | **397s**      |

---

> > ### Author Rebuttal · Reviewer_9nVe · 2026-04-08
> >
> > Thanks for your answer!! Yes I didn't read appendix and didn't notice they have already been mentioned there. I would encourage the authors to put some of the information into the main paper.

---

> > > ### Author Response · Authors · 2026-04-08
> > >
> > > We are truly delighted by the reviewer's follow-up and the kind decision to raise the score. It means a great deal to know that our rebuttal has fully addressed the reviewer's concerns, and we sincerely appreciate the careful and constructive feedback throughout the review process, which has been instrumental in strengthening our work. Following the reviewer's helpful suggestion, we will considering moving some details from the appendix into the main paper in the revised version, so that readers can more readily access the interaction protocol and computational cost analysis. We are sincerely grateful to the reviewer for the valuable time and insights that have helped refine our work.

---

### Decision · Program_Chairs · 2026-04-30

**Decision:**

Accept (regular)

**Comment:**

The paper proposes EmbodiedAct, a framework that transforms LLM-based scientific agents from passive "execute-then-observe" systems into active embodied agents with runtime perception and closed-loop control over simulation environments. All four reviewers converged on acceptance (scores 4, 4, 5, 5), with all concerns marked fully resolved after rebuttal. The main initial concerns (computational cost and whether gains come from additional trial-and-error) were addressed by controlled experiments showing equal interaction budgets across methods, along with a detailed cost analysis demonstrating that EmbodiedAct reduces wall-clock time by 53% and LLM tokens by 48% relative to CodeAct. The framework is well-motivated, clearly designed, comprehensively evaluated across multiple benchmarks and model backbones, and extends beyond MATLAB to Python-based environments. The authors should move the cost analysis and interaction protocol from the appendix into the main paper in the revision.